



# Investigation of cirrus clouds properties in the Tropical Tropopause Layer using high-altitude limb scanning near-IR spectroscopy during the NASA-ATTREX Experiment

Santo Fedele Colosimo[1], Nathaniel Brockway[1], Vijay Natraj[2], Robert Spurr[3], Klaus Pfeilsticker[4], Lisa Scalone[5], Max Spolaor[1], Sarah Woods[6], and Jochen Stutz[1]

[1]Department of Atmospheric and Oceanic Sciences, UCLA, Los Angeles, CA, USA
[2]Jet Propulsion Laboratory, Caltech, Pasadena, CA, USA
[3]RT Solutions Inc., Cambridge, MA 02138, USA
[4]Institute of Environmental Physics, University of Heidelberg, Heidelberg, Germany
[5]Springer Heidelberg, Heidelberg, Germany
[6]Stratton Park Engineering Inc., Boulder, CO, USA

**Correspondence:** S.F. Colosimo (fedele@atmos.ucla.edu)

**Abstract.** Tropical Tropopause Layer cirrus clouds and their radiative effects represent a major uncertainty in the evaluation of Earth's energy budget. High altitude aircraft offer an opportunity to provide observations at cirrus cloud altitudes, most commonly using in-situ measurements of ice particle optical properties and composition. In particular, remote sensing of scattering properties and near-IR ice water absorption in the limb can provide unique insights into thin and sub-visible cirrus clouds. Here

we present novel spectroscopic observations of path-averaged ice water absorptions onboard NASA's Global Hawk aircraft, during the Airborne Tropical TRopopause Experiment, which took place in 2011, 2013, and 2014. The University of California Los Angeles and University of Heidelberg mini-DOAS instrument provided multi-angle limb-scanning observations of scattered solar radiation in the near-IR (900–1726 nm), allowing the identification of ice and liquid water, $O_2$, $CO_2$ and $H_2O$. The VLIDORT-QS radiative transfer (RT) code was specifically developed for this study, and used to simulate high altitude

limb observations for varied cloud scenarios. We performed a comprehensive sensitivity study, developing a fundamental understanding of airborne near-IR limb observations of cirrus clouds. We identified two general distinct cases: a linear regime for optically thin clouds, where the ice absorption is proportional to ice water content (IWC), and a regime for optically thick cirrus clouds, where ice absorption is in saturation and independent of IWC. Results also demonstrate how molecular oxygen absorption can be used to infer information on optical properties of ice particles in the second regime only, with minimal

information for thin cirrus clouds. We also explored the feasibility to retrieve IWC from mini-DOAS path-averaged ice water absorption (SIWP) measurements. This innovative interpolation-based approach requires a small number of RT calculations per observation to determine the sensitivity of SIWP to IWC. Spectral retrievals were applied for a particularly interesting case during Science Flight 2 over Guam in February 2014, during which the aircraft flew in circles in the same general area for an extended period of time. Retrieved IWC results are consistent with independent in situ measurements from other instru-

ments on board. The measurements of ice particle scattering and absorption at different azimuths relative to the sun and at





different altitudes represents a unique opportunity to test our approach and to infer properties of the ice particles, together with information on cirrus cloud radiative transfer.

## 1  Introduction

Cirrus clouds are poorly understood components of the weather and climate system. Despite being present at every latitude, their occurrence increases from the polar to tropical regions [Sassen et al., 2008]. Cirrus clouds cover up to 43% of the tropics at any given time [Wylie and Menzel, 1999] and are frequently observed in the tropical tropopause layer (TTL), a transition zone between the tropical upper troposphere and the lower stratosphere (~13-18 km) [Fueglistaler et al., 2009]. In the TTL, thin and sub-visible cloud structures strongly influence the total energy budget, with a large impact on radiative forcing, and hence on Earth's climate [Jensen et al., 2013]. The impact on the radiative budget is twofold: in the solar region, they reflect light (albedo effect), thus leading to a negative forcing, while in the thermal infrared, they lead to a positive forcing (greenhouse effect) [Jensen et al., 1996; McFarquhar et al., 2000; Hartmann et al., 2001]. The balance between these two effects determines whether cirrus clouds cause a net positive or negative forcing. In particular, thin cirrus clouds have been shown to have a positive radiative forcing [Jensen and Toon, 1994], thus acting as important regulators of climate [Ramanathan and Collins, 1991; Randel and Jensen, 2013]. Due to the coexistence of convective (troposphere) and radiative (stratosphere) regimes in the TTL, the presence of cirrus clouds also regulates the stratospheric-tropospheric exchange, which influences both the water budget [e.g. Jensen and Toon, 1994; Fueglistaler et al., 2009] and the transport of precursors of reactive trace gases into the stratosphere [Fueglistaler et al., 2009; Aschmann et al., 2009; Aschmann et al., 2011; Brinckmann et al., 2012; Hossaini et al., 2012]. Finally, cirrus clouds can impact the chemical composition of the atmosphere through incorporating nitrogen and sulphate species as well as activating more reactive species, such as halogens [Popp et al., 2004; Popp et al., 2006; Borrmann et al., 1997; Solomon et al., 1997; Hobe et al., 2011; Abbatt et al., 2012; Bregman et al., 2002; Lowe and MacKenzie, 2008]. It is thus critical to obtain a detailed understanding of their physical and optical characteristics, as the radiative transfer (RT), water budget, and chemistry depend on their micro-physical properties, i.e., ice particle size distribution and habit, as well as their vertical location and horizontal structure. While cirrus clouds occur frequently in the TTL [Wylie and Menzel, 1999], they are often optically thin and thus difficult to observe, especially from space-borne remote sensing instruments [e.g. King et al., 1992; Rolland et al., 2000; Platnick et al., 2003; Wang et al., 2012]. Recently, airborne experiments have provided important information about the properties of these type of clouds thus improving space based observations [Wolf et al., 2017; Krisna et al., 2018]. However, data is still sparse and hence our understanding of thin cirrus clouds, especially at the low end of the optical thickness spectrum, remains weak. There is thus a crucial need for developing and applying sensitive methods to observe thin cirrus clouds for a better understanding of their physical and optical properties, to improve space-based observations, and to provide better information for global climate models.

In this study, we demonstrate the use of ice water absorption as a remote sensing parameter to study cirrus clouds. We use limb-scanning near-infrared (NIR) absorption spectroscopy measurements onboard NASA's Global Hawk (GH) unmanned aircraft vehicle, during the Airborne Tropical TRopopause Experiment (ATTREX) mission [Jensen et al., 2013], to derive





slant ice water paths (SIWP) (Section 2). A new RT code, capable of modeling limb geometries in the presence of scattering

particles is used to investigate how ice absorption is influenced by both radiative and environmental conditions (Section 2). After the validation of the capabilities of the RT model (RTM) by means of sensitivity tests (Section 3), we use our improved understanding of theoretical ice absorption limb observations (and the developed RT tools) to retrieve ice water paths (IWP) from ATTREX measurements (Section 4). We show that optical absorption measurements of ice are powerful tools to derive information on cirrus clouds and ice water path in the TTL.

## 2 ATTREX observations


The NASA ATTREX mission was an airborne experiment that performed a series of measurement campaigns with NASA's GH long range unmanned aircraft vehicle, to improve our understanding of the physical and chemical processes in the TTL [Jensen et al., 2013]. The payload of this experiment consisted of several remote and in situ instruments, specifically designed to provide information on water vapour, cloud particles properties, and trace gases, with a particular focus on the investigation

of the dehydration of air entering the stratosphere by ice crystal growth and sedimentation near the cold tropical tropopause. Measurements of IWC relied on the NOAA Water instrument, consisting of a tunable-diode laser absorption spectrometer, capable of measurements of upper troposphere/lower stratosphere water vapor and enhanced total water [Thornberry et al., 2015]. Information about the ice crystal size distributions, habits, and concentrations were provided by the Hawkeye instrument, consisting of a Fast Cloud Droplet Probe (FCDP), a two-dimensional stereo (2D-S) optical probe, and a cloud particle

imager (CPI) [Lawson et al., 2001, Lawson et al., 2006, Lawson, 2011, Lawson et al., 2017]. Accurate cloud height profiling was supplied by LIDAR images from the Cloud Physics Lidar instrument [McGill et al., 2002], a low-pulse-energy laser at high repetition rate, coupled with solid-state photon-counting detectors.

As part of this scientific effort, UCLA and the University of Heidelberg deployed a Mini Differential Optical Absorption Spectroscopy (MiniDOAS) instrument with the primary goal of studying bromine chemistry [Stutz et al., 2017, Werner et al., 2017],

using absorptions in the ultraviolet and visible spectral regions. As secondary goal was the investigation of ice absorptions in the NIR wavelength region as a tool to study cirrus clouds, which is the focus of this study.

### 2.1 MiniDOAS Instrument and Measurement

NIR spectra were recorded with the MiniDOAS instrument during the ATTREX missions in 2011, 2013, and 2014; for a detailed description of these experiments, the reader is referred to (Stutz et al., 2017). The NIR part of the instrument used a

scanner/telescope unit mounted at the portside of the GH with a viewing direction 1° along the flight direction. The telescope can measure at any angle between zenith and nadir, but was pointed near the limb (-0.5° elevation angle) to increase light-path lengths through cirrus clouds. The rectangular viewing opening angle was approximately $0.2° \times 1°$ for the vertical and horizontal dimension respectively. Real-time pitch data from the aircraft was used to maintain this angle to within $\pm 0.2°$ [Stutz et al., 2017]. At least once during each flight the telescope was directed onto a glass diffuser mounted in the zenith direction

to measure a direct solar spectrum. Light from the telescope was fed into an Ocean Optics NIRQuest512 Spectrometer via a



glass-fiber bundle. The spectrometer was mounted inside a vacuum-chamber and its temperature was maintained at $0°C$ via an ice water bath. Exposure times were determined with an auto-exposure algorithm that kept the detector signal at 50% of its maximum, with a maximum exposure time of 30 s. The total integration time was often on the order of 30 s from 900-1700 nm, on 512 pixels, and with a FWHM of $\approx 17$ nm. The measured spectra were offset, dark-current corrected, and divided

by the observed high-altitude solar spectrum to remove instrument structures and solar-spectrum signatures [Platt and Stutz, 2008]. These spectra were then analyzed with the DOASIS software package [Kraus, 2006] using the approach described in the following section .

## 2.2    Spectral analysis

The ice absorption retrieval was performed on the logarithm of the ratios of near-limb and direct-sun spectra:

$$S(\lambda) = ln\left[\frac{I_{limb}(\lambda)}{I_{sun}(\lambda)}\right] \tag{1}$$

Such a spectrum was then modeled using a function, F($\lambda$), comprising a linear combination of the various reference spectra, G$_j$($\lambda$) and a low order polynomial, P($\lambda$).

$$F(\lambda) = P(\lambda) + \sum a_j \times G_j(\lambda) \tag{2}$$

In classical DOAS [Platt and Stutz, 2008], the scaling factors $a_i$ and the polynomial $P(\lambda)$ are determined using a combination of a linear and non-linear least squares fitting algorithms minimizing the difference between S($\lambda$) and F($\lambda$). Factors $a_i$ are proportional to the concentrations of the gas absorbers. This is, however, not the case for $O_2$, $CO_2$, and water vapor, which have high-resolution absorption structures that are often in saturation. In addition, ATTREX variations in altitude, cloudiness, surface albedo, etc., can change column densities of $O_2$, $CO_2$, and water vapor by orders of magnitude. These large variations,

together with saturation effects, can lead to nonlinear changes in the depth of the absorption lines at low spectral resolution. The shape of the convolved absorption spectrum will thus depend on the column density [Goody and Yung, 1995].

To overcome this challenge, we have adopted an approach where a look-up table of simulated trace gas reference spectra is used to ensure that the fitting factors a$_j$ are close to 1. In this scenario, these fitting factors vary only in a linear regime that is proportional to the variation of the column density around the value at which the reference was simulated. Calculations

for spectra in the look-up table were based on high-resolution absorption cross sections for $O_2$, $CO_2$, and $H_2O$, taken from the HITRAN 2012 [Rothman et al., 2013] database, for a temperature of 200 K and a pressure of 0.1 atm, i.e. typical values encountered near the tropical tropopause. The assumption that all absorption occurs at high altitudes (ignoring absorption occurring in the lower part of the atmosphere), is partially justified for limb observations in the nearIR spectral range since contributions from scattering in the atmosphere is greatly reduced.

A Fast-Fourier Transform method was employed to convolve the high resolution intensity spectra over the range 894 nm to 1728 nm, with pure trace gas absorptions at a given column density, using a convolution kernel determined from the 1243.9 nm Ar emission line. Column densities were varied across the range expected during the ATTREX experiment.





| Parameter | $O_2$ Fit | $CO_2$ Fit | Ice Fit |
|---|---|---|---|
| Wavelength Range | 1160-1293 nm | 1508-1678 nm | 1338-1631 nm |
| Polynomial Order | 2 | 5 | 3 |
| References: | $O_2$ | $CO_2$ | 2 $CO_2$ |
| | $CO_2$ | | $H_2O$ |
| | | | Liquid Water |
| | | | Ice Water |
| | | | Ring Spectrum |

**Table 1.** *List of parameters implemented in the DOAS analysis fit routines.*

We have adopted an iterative approach to the first stage of the fitting, in which the result, $a_i$, of the first fit is used to select a gas reference spectrum from the pre-convolved look-up spectrum data set. A new fit is then performed and another set of pre-convolved reference spectra is chosen. This procedure is repeated until all trace gas fitting coefficients, $a_i$, are within 1% of unity. The second-stage and final fit is then performed to determine the $a_i$ factors used to calculate the column densities based on those used to simulate the final set of references.

Absorption cross-sections of ice and liquid water [Warren and Brandt, 2008] do not have narrow structures, and were included directly in the fitting process, with the resulting fitting factors then used to derive the ice and liquid water paths (see below). The retrieval routine determines the scaling factors $a_i$, and since we use an iterative approach, the path-integrated absorption is the product of $a_i$ (which is always close to 1) and the column density used to calculate the reference spectrum $G_j$.

To reduce spectral interference between trace gas absorption signatures, we used different spectral windows for the retrievals of ice, $O_2$, and $CO_2$, each with different model functions, i.e. different combinations of reference spectra (Table 1). The $CO_2$ reference spectra used in the ice fit were made by separating the reference for wavelengths lower than 1498 nm from the portion with wavelengths higher than 1498 nm. The latter portion is then constrained to the amount derived from the $CO_2$ fit routine. This is done to account for RT differences between the wavelength ranges when ice absorption is high. The Ring spectrum in this fit accounts for the effects of Fraunhofer "filling-in" interference signatures caused by rotational Raman scattering, and is based on [Bussemer, 1993]. A ring spectrum was found to be unnecessary for fitting $O_2$ and $CO_2$ (see table 1) due to the lack of narrow solar structures.

An example of a retrieval is shown in Fig. 1 panel (a), where gas/ice reference spectra (red lines) agree well with the retrieved spectral structures (black lines). The absorption by ice is the strongest spectral signature in this spectrum, with an optical depth of ~0.45. Optical densities of $O_2$ and $CO_2$, in the range of 0.03-0.04, are also retrieved with high accuracy.

A complication in the retrievals arises when low-lying liquid water clouds introduce strong water vapor and liquid water absorption [Segelstein, 1981]. Panel (b) in Fig. 1 shows an example with a thin liquid water cloud situated below a cirrus ice cloud during Science Flight 2 (SF2). The retrieval is still able to quantify all three phases of water, but a flattening of the intensity between 1350-1470 nm due to strong liquid and gaseous water absorption is evident. During parts of the ATTREX





**Figure 1.** *An example of DOAS optical density fitting, comparing the reference spectra (red lines) of $O_2$, $CO_{2,}$, $H_2O$, and ice, and the retrieved spectral structures (black lines) is shown in panel (a), (b), (c), and (d) respectively. A pseudo-saturated case, where strong gas and liquid water absorptions flattens the intensity (panel (e)) signal (shaded areas), making the ice retrieval difficult but still possible, is shown for water and ice, in panel (f), (g), and (h).*



flights, these gaseous and liquid water absorptions were even larger than shown here, and some spectra are saturated in this wavelength range. For these cases, we developed a second fit approach where this wavelength interval is omitted. Although this second fit is unable to retrieve water vapor and liquid water, it is still able to determine ice absorption, albeit with larger

errors. Due to additional difficulty interpreting these layered-cloud cases with our RT code, we will mostly focus on cases where liquid water absorption is small.

Our gas retrieval routine generates the slant column density (SCD), i.e., the integral of the trace gas concentration along a light path that reaches our instrument:

$$SCD = \int_0^l c(s) \cdot ds \tag{3}$$

where c(s) is the gas concentration along the light path. Since the fit must include a solar reference spectrum, which will contain a small amount of trace gas absorption from the atmosphere above the aircraft, the fitting calculates the differential SCD (DSCD), which is the difference between the limb-path and solar SCDs:

$$DSCD = SCD_{limb} - SCD_{sun} \tag{4}$$

As we use direct solar light in the absence of overhead clouds, we calculate $SCD_{sun}$ as an integral of the species along the

geometric path from the aircraft to the top of the atmosphere at the time of the solar observation, i.e., using the solar zenith angle (SZA) and flight altitude at the time of measurement. Gas profiles above the aircraft are based on the hypsometric approximation, combined with aircraft observations of temperature and pressure and a stratospheric lapse rate of $-2.5\,°C/km$. It should be noted that the contribution of $SCD_{sun}$ is typically less than 2% and 7% of $SCD_{limb}$ for $O_2$ and $CO_2$, respectively.

The ice absorption observations from our instrument can be interpreted in a similar way. The ice water path (IWP) is defined

as the vertical integral of ice mass concentration through a cloud, a quantity similar to the vertical column density of a trace gas. However, our instrument actually retrieves a slant ice water path (SIWP) defined as:

$$SIWP = \int \frac{M_{ice}}{\rho_{ice}} ds \tag{5}$$

where $M_{ice}$ is the mass concentration of ice along the light path $ds$. This equation is a theoretical explanation of what the SIWP represents, and also includes ice density $\rho_{ice}$ to reflect the fact that the absorption coefficient is in units of length ($cm^{-1}$) and,

consequently, we retrieve the ice light path in units of $\mu m$ (Fig. 2). It should be noted that the solar reference spectra used in the analysis are measured at the aircraft top altitude under cloud-free conditions, i.e., without ice between the instrument and the sun. Therefore, we do not report a differential SIWP, in contrast to gas retrieval.

Both the SCDs and the retrieval uncertainties span several orders of magnitude. The retrieval routine calculates an error for each spectrum and detected species. Based on these errors, we have determined that the minimum detection limit is twice the

quartile uncertainty, in order to remove the influence of higher SCD observations. This may give a better representation of the actual detection limits of the spectrometer. Although the detection limit changed with the observation scenario, the relative uncertainty remained relatively constant for $O_2$ (3%), $CO_2$ (8.2%), and Ice (7.6%). Unsaturated observations of liquid water



**Figure 2.** *Panel (a),(b), and (c) show the SIWP, $O_2$ and $CO_2$ SCD from the first leg of ATTREX SF2 over the ocean near Guam, on February 16, 2014. Panel (f), (g), and (h)) show the same parameters for the last portion of the flight, on February 17, 2014. Water vapor and liquid absorption were low and are not shown here. Panel (d) and (e), and panel (i) and (l), show the flight altitude and the relative azimuth between the solar and viewing directions, for the two portions of the flight respectively. Error bars are similar in size to the symbols and are therefore not included.*



and gaseous $H_2O$ had a 10% and 4.9% uncertainty respectively. Consequently, we quote three different retrieval detection limits (Table 2): the best achievable detection limit, the median detection limit at conditions where liquid water absorption is not saturated, and the median detection limit under saturated conditions.

| Species | Best | Unsaturated | Saturated |
|---|---|---|---|
| $O_2$ (molec. cm$^{-2}$) | $2.14\times10^{24}$ | $3.77\times10^{24}$ | $1.04\times10^{25}$ |
| $CO_2$ (molec. cm$^{-2}$) | $3.03\times10^{21}$ | $5.17\times10^{21}$ | $7.45\times10^{21}$ |
| SIWP ($\mu$m) | 3.3 | 3.6 | 17.8 |
| $H_2O$ (molec. cm$^{-2}$) | $2.08\times10^{19}$ | $2.43\times10^{19}$ | - |
| $H_2O$ (liquid) ($\mu$m) | 6.6 | 8.4 | - |

**Table 2.** *Detection limits based on $2\sigma$ fit errors.*


## 2.3 ATTREX SF2: a case study

For this study, we focus on a portion of SF2 on February, 16$^{th}$ 2014. This flight took place over Guam, a U.S. island territory in Micronesia in the Western Pacific, located at (13°N,144°E), and was characterized by cirrus cloud measurements in a limited area for an extended period of time. This was a uniquely useful flight for the purpose of this study, since the Global Hawk flew
rectangular vertical spirals with steep ascending/descending manoeuvres, allowing rapid measurements to be taken at different viewing directions and altitudes. Retrievals from SF2 are shown in panels (a) and (b) in Fig. 2, where the change in viewing direction is indicated as the change in the relative azimuth angle between the mini-DOAS viewing direction and the sun (with 0° and 360° indicating that the instrument looks towards the sun, and 180° denoting the sun behind the instrument). LIDAR and in situ data reveal that, during the ascents and descents (altitude plot in Fig. 2), the aircraft frequently passed through cirrus
clouds. The graphs show how $O_2$ and $CO_2$ SCDs behaved similarly, confirming our expectation that, since their concentration profiles are constant in time and of similar shape, their behavior is controlled by RT effects. This is also evident from how the SCDs changed with azimuth angle One conclusion from this observation is that it is sufficient to consider only one of these gases to constrain the RT model. As $O_2$ retrievals are more accurate and stable, we have focused our analysis on this species. The retrieved SIWP is in the range of 0-130 $\mu$m, showing that the observed light has been impacted by ice particles at all
altitudes. The SIWP behavior is also dependent on azimuth, and SIWP tracks the $O_2$ SCD. In fact, the data retrieved before 23:00 UTC show a reasonably high correlation between the ice absorption and log of $O_2$ SCD (Pearson Correlation Coefficient of 0.67). This is somewhat counter-intuitive, considering that an increase in ice particle concentration would increase the amount of scattering and thus shorten the $O_2$ absorption path in the limb. On the other hand, part of the light observed by the instrument originates from below the cloud, where $O_2$ concentrations are much higher, and enhanced scattering could increase
the amount of light from below and thus lead to an increase in $O_2$. This observation is revisited in Section 3.
If we focus on the last part of SF2 (panel (b) in Fig. 2), we notice that for most of this flight section the SIWP and gas SCDs show the same behavior as that for the first part of SF2 (panel (a) in Fig. 2). However, the period between 5:20 UTC and 5:50





UTC is clearly different. As the Global Hawk climbs above 17 km altitude, around 5:20 UTC, the SIWP drops to near-zero

values. At the same time $O_2$ and $CO_2$ SCDs also drop to very low values. We interpret this behavior to arise from observations

above a thick cirrus cloud, where no or little ice particles are within the observation volume. The remaining non-zero SIWP

comes from upwelling light from the cirrus cloud that is then scattered by air molecules into the line of sight (LOS) of the

instrument. The cloud also seems to block light from lower parts of the atmosphere, so that the gas SCDs arise solely from

absorption within the (limb) line of sight and from above. The subsequent descent into this thick cloud also leads to the highest

SIWP observed in this flight. The peak in SIWP is accompanied by an anti-correlated decrease in $O_2$ and $CO_2$ SCDs, likely

caused by the fact that the cirrus cloud is optically very thick and most of the observed light originates from scattering by ice

particles rather than from air molecules. Again, no light from the lower atmosphere reaches the instrument.

These examples clearly show that measured $O_2$ and ice absorption are impacted by cirrus cloud geometry, solar zenith angle,

viewing zenith angle, azimuth angle, flight altitude, and other geometrical and physical factors. This motivates the need for RT

model calculations to properly interpret ice absorption measurements under a variety of different conditions. As part of this

study we therefore developed an expanded version of the VLIDORT radiative transfer code [Spurr, 2006], capable of handling

limb geometries.

### 2.4 Radiative Transfer Model

The VLIDORT-QuasiSpherical (VLIDORT-QS) vector RT model [Spurr et al., 2022], was used to simulate the ATTREX observations and to perform the sensitivity tests. Based on the recent version 2.7 (released early 2015) of the VLIDORT code [Spurr,

2006], the completely new VLIDORT-QS code was specifically designed and developed for the interpretation of high-altitude

limb scanning measurements. Ray tracing in a fully spherical atmosphere is a crucial capability for the simulation of ATTREX

measurements, which are made in the limb at 14-20 km altitude with absorption path lengths of up to 300 km. To account for

various viewing angles (the up, down, or limb viewing regimes), the model must be able to perform RT calculations along all

fully illuminated (no shadow or twilight conditions are currently considered) line-of-sight (LOS) paths.

The Earth's atmosphere is quite strongly curved. In order to obtain sufficient accuracy for scenarios such as those for ATTREX,

where long paths through the atmosphere are present, it is necessary to perform RT in a curved spherical-shell atmosphere both

for the single scatter (SS) and the multiple scatter (MS) contributions to the radiation field. To this end, VLIDORT-QS relies

on a spherical-shell optical model, comprising a series of optically uniform layers extending from the surface to the TOA with

resolution 1-2 km in the troposphere and 0.5 km in the upper troposphere/lower stratosphere, where the ATTREX measure-

ments were taken. VLIDORT-QS assumes horizontal uniformity of the atmosphere. For the SS component, VLIDORT-QS

solves the RT equation (RTE) accurately for each path segment in a curved shell atmosphere, for any of the regimes noted

above. A given segment SS source term is established by integrating the SS RTE numerically from segment start to finish,

using a Gauss-Legendre quadrature scheme defined by $N_j$ points of abscissa and weights $\{\alpha_j, \chi_j\}$ $(j = 1, 2, ..., N_j)$, based on

the change of LOS zenith angle $\alpha$. These segment quadrature points define an intermediate set of solar angles $\{\theta_j\}$, for which

solar-beam ray-tracing is performed, plus a corresponding set of vertical optical thicknesses $\{\tau_j\}$. Once the layer SS source

terms are found, the radiation field at the observer's position is developed recursively, starting with the value at the path origin





(TOA or surface) and using the cumulative LOS transmittances to propagate these source terms. Path segments are mostly defined by layer boundaries, except for those segments in the limb view on either side of the tangent point. If the observer is placed between two layer boundaries, then the final segment of the LOS will encompass a partial layer.

For the MS component, the code relies on an adapted form of the VLIDORT RT model in which a single call to VLIDORT, for each segment, will generate a set of plane-parallel discrete-ordinate solutions to the RTE at each of quadrature points $\{\alpha_{sj}, \chi_{sj}\}$ $(j = 1, 2, ..., N_j$ in that segment). The MS segment source term is then derived by integrating these MS solutions using the same quadrature scheme as was used for the SS field. In essence, the plane-parallel source function integration scheme normally employed in VLIDORT is replaced by a scheme done in full spherical geometry. This is in some ways equivalent to

a piece-wise concatenation of MS 1-D RT solutions in a 3-D scenario. VLIDORT-QS is not a truly spherical model; it cannot deal with earth-shadow scenarios and is confined to geometrical configurations in which all points on the LOS path are fully illuminated (no incident-beam twilight situations, solar zenith angles must be less than 90 degrees). However, all MS solutions from VLIDORT are obtained in the pseudo-spherical approximation, in which solar beam attenuation (before scattering) is always treated accurately for a curved atmosphere.

An accurate and comprehensive theoretical description of the new VLIDORT-QS RT code is provided in [Spurr et al., 2022], where the RT code is validated against the standard VLIDORT model for nadir-view (low sphericity) scenarios, and against the fully spherical Monte Carlo MYSTIC model [Emde et al., 2016] (part of LibRadTran RT code package [Mayer and Kylling, 2005]) for cases with higher sphericity. The validation includes general near-IR simulations for high-altitude aircraft limb/nadir viewing scenarios, including a Rayleigh-only regime, and in the presence of stratospheric clouds, for different observational

altitudes, viewing angles, and surface albedo. Results of these tests showed average differences among the RT codes on the order of a few percent, depending on the geometry and the scenario.

## 2.5 ATTREX RT calculations

In order to simulate realistic cirrus-cloud environments, we replicated the general conditions of the ATTREX scenarios, as

encountered on the February 16, 2014 SF2. The atmosphere was divided into 50 layers, with a pressure range from 1015.3 mbar (surface) to 9.4 mbar (TOA), and an altitude range between 0 and 32 km. The vertical grid spacing was set to 1 km between 0-10 km, and 0.5 km above 10 km altitude. Temperature, pressure, molecular and scattering particle properties are considered to be homogeneous within each layer. In the near-IR spectral range (900-1800 nm), the main atmospheric absorbers are $O_2$, $CO_2$, and $H_2O$. Because of the low atmospheric pressure due to the measurements altitude, the quadratic contribution

at 1067 nm and 1270 nm due to the absorption induced by the $O_2$-$O_2$ collision [Smith and Newnham, 1999], has not been considered in this study. Vertical profiles of $O_2$ and $CO_2$ are calculated using fixed mixing ratios of 0.21 ppm and 370 ppm, respectively. The surface is assumed to be Lambertian. No lower atmospheric clouds were considered in the model. We used a flight altitude of 16.5 km for our sensitivity tests, unless otherwise noted, and the true aircraft altitude for retrievals. The ice cloud deck was located between 14 and 18 km for all tests.

The ice particle optical properties were determined from averaged in situ data provided by the CPI and FCDP instruments





(described in Section 2) as part of the Global Hawk payload. Images from the CPI instrument provided information about the habit classification, while FCDP measurements yielded the number density of the particles. It is worth notice that the FCDP instrument measures particles between 1 - 50 microns, defining the size interval for the microphysical properties used in the ice retrievals from the DOAS observations.

During SF2, the aircraft sampled the same area for an extended period of time. However, ice particle numbers varied widely. We therefore calculated an average over the period of interest for this study, using the ice volume related to the averaged particle number concentration $n$, the size distribution and the density of ice $\rho_{ice} = 0.91 \times 10^6$ g m$^{-3}$. This resulted in a nominal ice concentration of N$_{ice} = 1.22 \times 10^{-3}$ g m$^{-3}$. If not otherwise specified, $n$ is referred to as the nominal case in the rest of the paper. This value represents the baseline, and different ice concentrations are expressed as multiplicative factors with respect

to the nominal value.

Although classical Mie theory is not ideally suited for deriving optical properties of clouds with non-spherical ice particles, images from the Hawkeye-CPI data showed that the ice particles were largely quasi-spherical for the smaller particles size range covered by the FCDP instrument [Woods et al., 2018], and we have accordingly used Mie theory for the calculation of these properties. Furthermore, as will become clear in the subsequent discussion, an overall understanding of the principles of

limb ice absorption measurements is not in general impacted by the shape of the particles.

Since scattering from ice particles enhances simulated limb radiances (making the intensity in the absence of ice actually smaller than that in the presence of ice), a differential approach for the ice absorption simulation was implemented, wherein the ice absorption was evaluated using the intensity at two different wavelengths $\lambda_1$ and $\lambda_2$ according to:

$$\tau_{ice} = ln\left[\frac{I^0_{\lambda 1}}{I^0_{\lambda 2}}\right] - ln\left[\frac{I_{\lambda 1}}{I_{\lambda 2}}\right] \tag{6}$$

where $I^0_{\lambda 1}$, $I^0_{\lambda 2}$ and $I_{\lambda 1}$, $I_{\lambda 2}$, indicates the intensities in the absence and presence of ice for the two selected wavelengths, respectively. It should be noted that this approach is similar to that underlying the differential absorption retrieval described above. Wavelengths of 1548.1 nm and 1550.3 nm, at the edges of a broad ice absorption feature, were selected for $\lambda_1$ and $\lambda_2$ respectively. Gas absorption lines are weak at these wavelengths, thus reducing any effect that gas absorption has on ice particle scattering. Oxygen absorption, on the other hand, was calculated according to Beer's law

$$\tau_{O_2} = \ln\left[\frac{I^0_\lambda}{I_\lambda}\right] \tag{7}$$

where gaseous optical absorption is evaluated as the logarithmic ratio between intensities calculated with and without the absorber, $I_\lambda$ and $I^0_\lambda$ respectively. In this study, oxygen absorption was calculated at $\lambda$ = 1260.8 nm, using two consecutive runs of the model (with and without oxygen absorption).

## 3 Sensitivity studies for ice absorption in cirrus clouds

Observations during the ATTREX experiment revealed that both ice and oxygen absorptions are highly variable, not only due to the presence of varying amounts of ice, but also due to different solar geometries, altitudes, etc. The first goal of this study





was therefore to develop a fundamental understanding of the mechanism and dependencies of ice scattering/absorption in cirrus clouds under near-limb viewing conditions, using sensitivity studies for ice and oxygen absorption under various conditions (Table 3). For each sensitivity test, the ice concentration was varied by scaling the nominal concentration over five orders of

| Test | Parameter | | | |
|---|---|---|---|---|
| | Solar Zenith Angle | Solar Azimuth Angle | Surface Albedo | Cloud Height [km] |
| Geometry | 20°, 45°, 70° | 0°, 45°, 90°, 135°, 180° | 0.05 | 14-18 |
| Surface Emissivity | 45° | 45° | 0.1, 0.2, 0.4, 0.6, 0.8 | 14-18 |
| Cloud Altitude | 45° | 45° | 0.05 | 14-17, 15-16, 15-17, 16-17, 17-18 |

**Table 3.** *The table lists the tests performed in this study, with the relative variation of the parameter of interest. The flight altitude is set to 16.5 km for all tests.*


magnitude, representative of typical ice concentrations from thin to thick cirrus clouds [Krämer et al., 2016]. All tests showed a similar general behavior as the ice concentration was varied. Ice absorbance, i.e. SIWP, shows a linear dependence on ice concentration for scaling factors below $n \times 10^{-1}$ (Fig. 3, panel (a), panel (c), and panel (e)). Oxygen absorbance remains constant in this range (Fig. 3, panel (b), panel (d), and panel (f)). For ice concentrations above the nominal $n$, ice absorbance

begins to flatten, while oxygen begins to deviate from its constant value at low concentrations below $n$. This transition depends on the solar geometry as will be discussed in further detail below.

To study the transition from a linear to a non-linear ice absorption regime, we separated the SS and MS contributions to the model output intensities (Fig. 4). It should be noted here that the MS contribution in VLIDORT-QS includes radiation diffusely reflected by the surface and the atmosphere below the aircraft. The low ice concentration regime is described by a constant

MS/SS intensity ratio. This ratio is non-zero due to the surface scattering/reflection contribution. MS becomes more important at the transition point. However, the MS/SS ratio remains below 2%, indicating that MS only plays a minor role in the flattening of the ice absorption signature and that SS dominates overall.

The linear dependence of ice absorbance is thus a consequence of the increased scattered radiance as a function of the greater number of particles. The lack of sensitivity of oxygen absorbance at low ice concentrations can be explained by the small

contribution that ice particle scattering has to the overall radiance compared to scattering by air molecules in this regime. In the transition regime, the ice particle scattering contribution increases, thus making the gas absorbance more sensitive to lower parts of the atmosphere, ice particle optical properties, and sun geometry.

To better understand the impact of solar geometry, we simulated ice and $O_2$ absorption for five different azimuth angles (SAZ: 0°, 45°, 90°, 135°, 180°) and for three solar zenith angles (SZA: 20°, 45°, 70°). The basic shape of the ice absorption curves

remains similar for all sun geometries (Fig. 3). The ice concentration at which the transition from linear to non-linear behavior occurs, and the slope in the non-linear regime, both depend on SAZ and SZA. In the low ice concentration regime, larger ice absorption occurs at lower SAZ (viewing towards the sun), while at larger SAZ there is a systematic decrease in the ice absorption, reaching a minimum when the sun is opposite to the instrument LOS. Interestingly, this pattern is reversed at higher



**Figure 3.** *Ice absorbance (left column) and O$_2$ absorbance at five azimuth angles (0°, 45°, 90°, 135°, and 180°) and for three different solar zenith angles (20°, 45°, and 70°), as a function of ice particle concentration.*

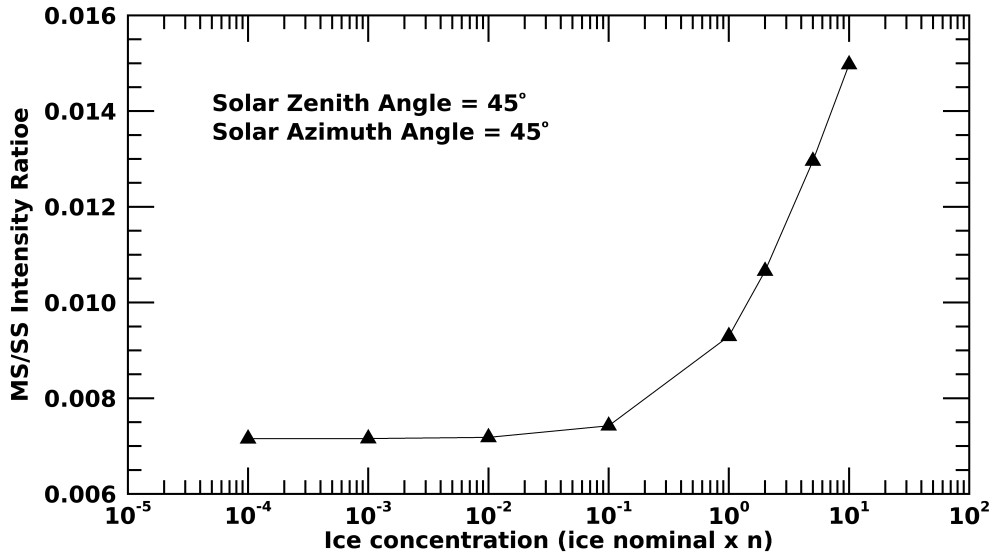

**Figure 4.** *Ratio of the contribution of SS and MS intensities as a function of ice concentration.*

ice concentrations: lower ice absorption is now associated with smaller SAZ, and an increase in SAZ returns to higher ice absorption.

This effect can be partially explained by strong forward scattering of the ice particles. Radiation is predominantly scattered along the direction of the incoming beam, while most of the absorption occurs along the instrument LOS. Conversely, back-scattered radiation observed at higher SAZ provides less absorption in the direction of the instrument LOS. The reversal at higher ice concentrations is likely due to more complex radiative effects, such as a higher contribution of the observed skylight radiance stemming from ice compared to air or the increased contribution of MS. The dependence on SZA is particularly evident for SZA=20° (panel (a) in Fig. 3), where ice absorption differences are reduced by the small SZA (no azimuthal dependence of the ice absorption at the zenith). At SZA=70° (panel (e) in Fig. 3), the longer atmospheric absorption path makes the effect of the SAZ more pronounced. The $O_2$ absorption behavior, for the same change in observational geometry, shows that at low ice concentrations the absorption is basically constant, while it diverges in the high ice concentration regime. The more complex response of $O_2$ absorbance at high ice concentration is explored in more detail below.

Limb measurements at high altitudes can be affected by radiation reflected by the surface. Consequently, a sensitivity test was performed for different surface reflectivities, in order to evaluate the influence of the surface albedo on the measurements (Fig. 5). We varied the albedo from 0.1 to an unrealistic value of 0.8 to demonstrate clearly the effect of the surface. Radiation from the surface contributes an additional term to the signal, proportional to the percentage of light reflected back into the atmosphere. Both ice and oxygen absorbances increase linearly with the albedo, showing this scaling effect.



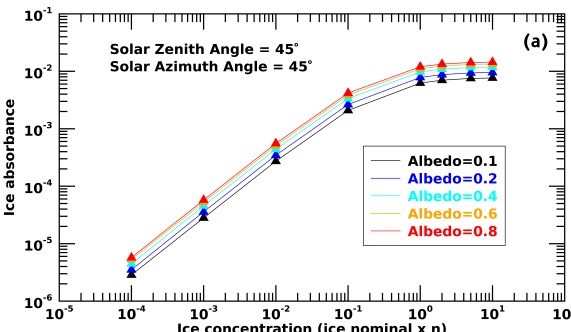
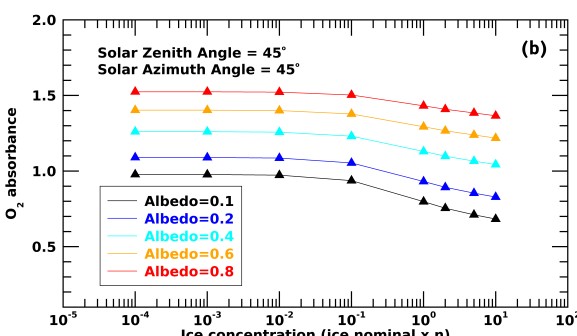

**Figure 5.** *Ice (panel (a)) and $O_2$ (panel (b)) absorbance as a function of ice concentration for four different surface albedos (0.1. 0.2, 0.4, 0.6, 0.8). Simulations were performed for SZA=45° and SAZ=45°.*

In order to investigate the effect of different cloud altitudes on the signal, we compared simulations for five different cloud geometries (14-17 km, 15-16 km, 15-17 km, 16-17 km, and 17-18 km), keeping the flight altitude constant at 16.5 km. This allowed us to simulate conditions close to those for the actual measurements, whereby the instrument is flying below, inside,
and above ice clouds (Fig. 6). Simulations of ice absorbance for observations inside and below the cloud behave similarly, while measurements above the cloud resulted in much lower ice absorbance. In the latter case, the instrument collects photons coming mainly from the thin atmosphere above the cloud. When the aircraft is below the cloud (17-18 km case in Fig. 6), most of the signal originates from the atmosphere below the aircraft, but solar radiation first passes through the cloud and is then scattered below it, contributing to a similar pattern present in observations taken inside the cloud.

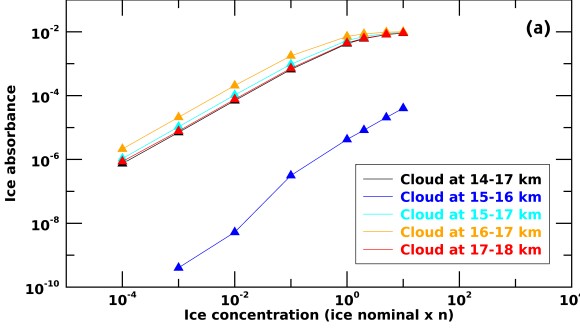
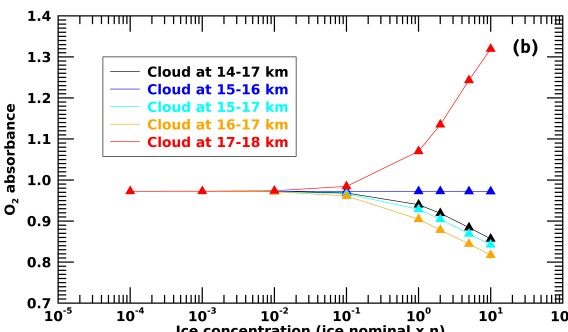

**Figure 6.** *Cloud altitude sensitivity simulations of Ice (panel (a)) and $O_2$ (panel (b)) absorbance, for 5 different cases: 14-17 km, 15-16 km, 15-17 km, 16-17 km, and 17-18 km, to simulate measurements above, inside, and below the cloud deck. Flight altitude is fixed at 16.5 km.*

The oxygen absorbance behaves as expected (Fig. 6), with no dependence on cloud geometry at low ice concentrations. At higher ice concentrations, when the instrument observes above the cloud (cloud located between 15-16 km), the oxygen absorbance remains constant, as photons only originate from above the cloud. Simulations for observation inside the cloud show


that the oxygen absorbance decreases with ice concentration as the oxygen absorption path shortens. When the aircraft is below

the cloud (located at 17-18 km), an increase in ice concentration (thicker cloud) enhances the signal as a consequence of an

increased contribution of the oxygen absorption in the lower part of the atmosphere.

### 3.1    Conceptual model of high altitude limb ice and oxygen absorptions

These sensitivity studies reveal a number of common features in limb observations of SIWP (ice absorbance) and $O_2$ SCD in

cirrus clouds:

–  At low ice concentration, SIWP is a linear function of ice concentration, while oxygen SCD remains constant

–  At high ice concentrations, SIWP levels off, while oxygen SCD decreases or increases depending on viewing geometry

–  SIWP and $O_2$ SCD are functions of viewing geometry. In particular the $O_2$ SCD at high ice concentration is highly

dependent on the viewing azimuth angle

–  SIWP and $O_2$ SCD are both functions of surface (and lower clouds) albedo

–  SIWP and $O_2$ SCD are both functions of cloud geometry and relative observation altitude

The first four items in this list allow us to develop a general understanding of our observations. To this end, we have used a

simplified description of the source of light observed by the limb-viewing mini-DOAS system. Despite their intrinsic difference,

the terms optical depth and absorption are considered synonymous throughout the paper, because both relate to $O_2$ SCD and

SIWP.

As illustrated in Fig. 7, the observed skylight radiance can be approximated by the sum of four different components. $I_S^R$ is the

radiance of sunlight scattered by air molecules. $I_S^M$ is the radiance of sunlight scattered by ice particles. $I_A^R$ and $I_A^M$ are the

radiances of upwelling light, i.e., sunlight scattered upward by the surface and the atmosphere (air molecules and ice particles),

respectively. These quantities can be written as:

$$I_S^R(\lambda) = I_S(\lambda) \cdot P_S^R(\theta, \vartheta, \lambda) \cdot \epsilon_R(\lambda) \cdot C_{air} \tag{8}$$

$$I_S^M(\lambda) = I_S(\lambda) \cdot P_S^M(\theta, \vartheta, \lambda) \cdot \epsilon_M(\lambda) \cdot N_{ice} \tag{9}$$

$$I_A^R(\lambda) = I_S(\lambda) \cdot R(\lambda) \cdot P_A^R(\theta, \vartheta, \lambda) \cdot \epsilon_R(\lambda) \cdot C_{air} \tag{10}$$

$$I_A^M(\lambda) = I_S(\lambda) \cdot R(\lambda) \cdot P_A^M(\theta, \vartheta, \lambda) \cdot \epsilon_M(\lambda) \cdot N_{ice} \tag{11}$$

Here $I_S(\lambda)$ represents the incoming solar intensity, $P(\theta, \vartheta, \lambda)$ represents the phase function for scattering events, $R(\lambda)$ is a

factor taking into account the radiation reflected by the surface/atmosphere below the aircraft, $\epsilon_R(\lambda)$ and $\epsilon_M(\lambda)$ represent the

Rayleigh and Mie extinctions respectively, $C_{air}$ is the concentration of air molecules, and $N_{ice}$ identifies the ice particle number

density. Using the following representation of the observed radiance

$$I_{obs}(\lambda) = I_S^R(\lambda) + I_S^M(\lambda) + I_A^R(\lambda) + I_A^M(\lambda) \tag{12}$$





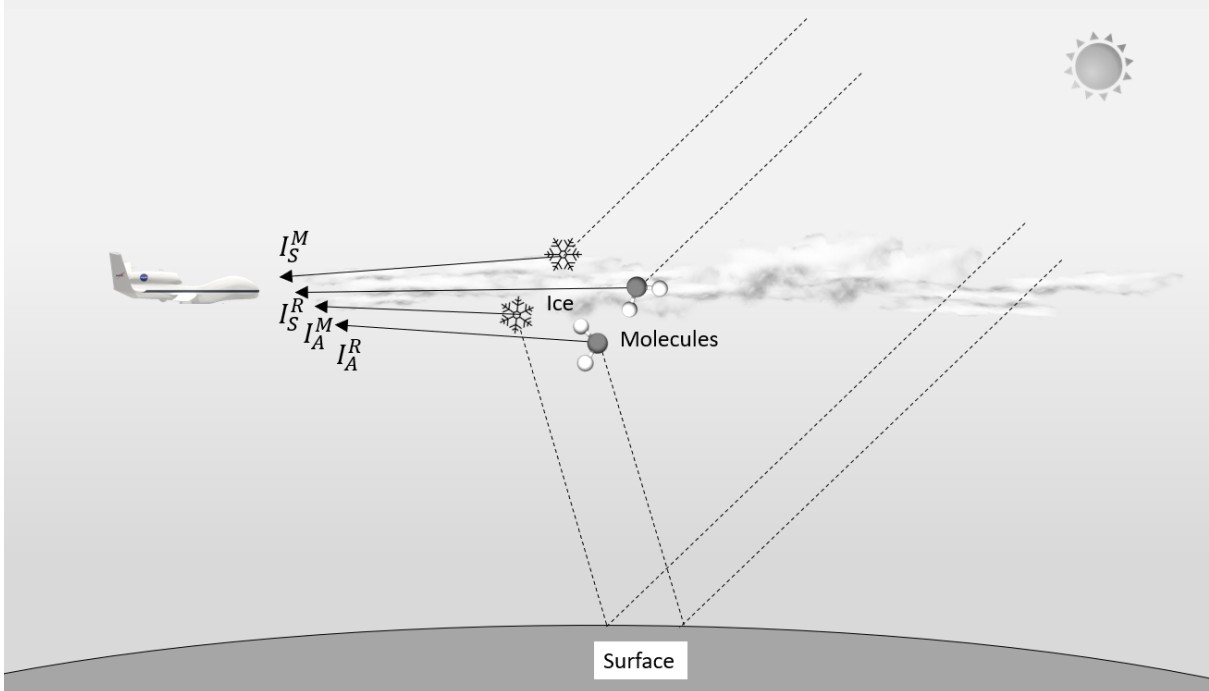

**Figure 7.** *Sketch of the radiative quantities defining the observed radiance:* $I_S^R$ *is the component of sunlight scattered by air molecules,* $I_S^M$ *is the component of sunlight scattered by ice particles,* $I_A^R$ *and* $I_A^M$ *are the radiances of upwelling light, after the interaction with the surface, due to molecules and particles, respectively.*

and remembering Eq.(7), we can now use these equations to describe the generic optical depth at wavelength $\lambda$ as

$$\tau_\lambda = ln\left[\left(P_S^R(\lambda) + R(\lambda)\cdot P_A^R(\lambda)\right)\cdot\epsilon_R(\lambda)\cdot C_{air} + \left(P_S^M(\lambda) + R(\lambda)\cdot P_A^M(\lambda)\right)\cdot\epsilon_M(\lambda)\cdot N_{ice}\right]$$

$$= ln\left[A\cdot\epsilon_R(\lambda)\cdot C_{air} + B\cdot\epsilon_M(\lambda)\cdot N_{ice}\right]$$

$$= ln(A\cdot\epsilon_R(\lambda)\cdot C_{air}) + ln\left[1 + \frac{B\cdot\epsilon_M(\lambda)\cdot N_{ice}}{A\cdot\epsilon_R(\lambda)\cdot C_{air}}\right] \tag{13}$$

where we have introduced $A = P_S^R(\lambda) + R(\lambda)\cdot P_A^R(\lambda)$ and $B = P_S^M(\lambda) + R(\lambda)\cdot P_A^M(\lambda)$ to simplify the equation. $A$ and $B$

are functions of the geometry, the radiation reflected by the surface/atmosphere below the aircraft, and the scattering properties of air (function $A$) and ice particles (function $B$), respectively. These general expressions for the optical depth can be approximated for two special cases (low and high ice concentrations) for oxygen and ice, depending on the different relative weights of the two terms in Eq.(13):

Case I: Low ice concentration

$$A\cdot\epsilon_R(\lambda)\cdot C_{air} >> B\cdot\epsilon_M(\lambda)\cdot N_{ice} \tag{14}$$





In this case, invoking $ln(1+x) \approx x$ for $x \approx 0$, Eq.(14) can be approximated as

$$\tau_\lambda \approx ln(A \cdot \epsilon_R(\lambda) \cdot C_{air}) + \left( \frac{B \cdot \epsilon_M(\lambda)}{A \cdot \epsilon_R(\lambda) \cdot C_{air}} \right) \cdot N_{ice} \tag{15}$$

The ice optical depth is calculated as a differential absorption (Eq.(6)) at two adjacent wavelengths $\lambda_1$ and $\lambda_2$

$$\tau_{ice} = ln\left[ \frac{I^0_{\lambda 1}}{I^0_{\lambda 2}} \right] - ln\left[ \frac{I_{\lambda 1}}{I_{\lambda 2}} \right] = \tau_{\lambda 1} - \tau_{\lambda 2} \tag{16}$$

The proximity of the chosen wavelengths allows us to take $A(\lambda_1) \approx A(\lambda_2)$, and we can assume further that $\epsilon_M(\lambda_1) \approx \epsilon_M(\lambda_2)$ because of the relatively weak dependence of Rayleigh extinction on wavelength in the ice absorption range. The combination of Eq.(15) and Eq.(16) leads to a compact expression for ice absorption for low ice concentration:

$$\tau^{low}_{ice} \approx \frac{B(\lambda_1) \cdot \epsilon_M(\lambda_1) - B(\lambda_2) \cdot \epsilon_M(\lambda_1)}{A \cdot \epsilon_R \cdot C_{air}} \cdot N_{ice} \tag{17}$$

Equation (17) clearly demonstrates that under conditions where scattering by air molecules outweighs that by ice particles, a linear dependence of ice absorption (SIWP) on the ice concentration is expected, with a proportionality factor that depends on the ice scattering phase function.

This result now raises the question why the oxygen absorption seems independent of $N_{ice}$ in our sensitivity tests. For the $O_2$ absorption, Eq.(15) can be simply approximated as

$$\tau^{low}_{O_2} \approx ln(A \cdot \epsilon_R(\lambda) \cdot C_{air}) \tag{18}$$

, which explains the lack of sensitivity of oxygen absorbance at low ice concentrations. In this expression, the function $A$ is large, and is independent of $N_{ice}$.

Equation (17) and Eq.(18) explain the behavior of $O_2$ and ice absorption in our RTM sensitivity tests at low ice concentrations and thus provide a conceptual model for this regime.


Case II: High ice concentration

$$\boxed{A \cdot \epsilon_R(\lambda) \cdot C_{air} << B \cdot \epsilon_M(\lambda) \cdot N_{ice}} \tag{19}$$

In this second special case, all of the previous approximations are no longer valid, and the optical depth is described again by Eq.(13) which, after combining with Eq.(16), leads to a new expression for the ice absorption:

$$\tau^{high}_{ice} \approx ln\left[ \frac{A \cdot \epsilon_R \cdot C_{air} + B(\lambda_1) \cdot \epsilon_M(\lambda_1) \cdot N_{ice}}{A \cdot \epsilon_R \cdot C_{air} + B(\lambda_2) \cdot \epsilon_M(\lambda_2) \cdot N_{ice}} \right] \propto ln\left[ \frac{B(\lambda_1) \cdot \epsilon_M(\lambda_1)}{B(\lambda_2) \cdot \epsilon_M(\lambda_2)} \right] \tag{20}$$

This explains why SIWP becomes independent of ice concentration $N_{ice}$, but at the same time remains strongly dependent on the viewing geometry and ice phase function, which is included in $B(\lambda)$. The factor $B(\lambda)$ also contains the factor $R$ (Eq.(10) and Eq.(11)), which includes the surface albedo. Consequently, ice absorbance at high ice concentrations depends in a complex way on the sun geometry and the phase functions at different wavelengths, as well as the atmosphere/surface below the cloud.





Further exploration of this regime thus requires detailed simulations with a proper RTM.

For oxygen absorption, Eq.(6) is unable to fully describe the behavior seen in Fig. 3, although the strong dependence on the viewing geometry, ice phase function, and atmosphere/surface through the term $B(\lambda)$, can partially explain the results of our RTM sensitivity tests.

## 4 Interpretation of ATTREX data

Using the improved understanding of our measurement technique, we have developed a procedure to retrieve ice concentration in cirrus clouds based on the SIWP as measured by the mini-DOAS instrument. This procedure was applied to a portion of SF2 (February 17, 2014, from roughly 05:00 am UTC to 8:30 am UTC), which offered the best opportunity to test and demonstrate our retrieval, thanks to changing viewing geometry and aircraft altitudes in the presence of cirrus clouds.

### 4.1 Radiative Transfer Calculations

For each mini-DOAS observation we performed three RT calculations to simulate ice absorptions. The model was constrained by in situ parameters/observations that are specific to our measurement and always available at high accuracy: these are the viewing and solar geometry, solar zenith and relative azimuth angles, viewing elevation, aircraft altitude, temperature and pressure. Other parameters needed to constrain the model are less certain, because their values observed on the NASA-GH 435 may not be representative for the extended volume probed by our instrument, or because measurements were not available. These parameters include the ice particle size distribution, the cloud height and depth, and the surface albedo. For the latter parameters we use a priori data from averaged ATTREX observations. Table 4 shows the ranges of the parameters used for the RT calculations for the portion of the flight considered in this study.

| Flight Altitude [km] | Solar Zenith Angle | Solar Azimuth Angle | Surface Albedo | Size Distribution | Single Scattering Albedo |
|---|---|---|---|---|---|
| 14.40-17.71 | 51.81°-86.15° | 0.20°-346.8° | 0.076-0.25 | Bimodal 3-5$\mu$m | 0.982-0.999 |

**Table 4.** *Actual parameter ranges used for the RT calculation, as provided by the on-board housekeeping data (except particle size distribution and single scattering albedo). The instrument viewing angle was set to -0.5°, relative to the limb (limb at 0°) for the entire portion of the flight considered in this study.*

The model was set up in the same way as that described in section 2.5. Cloud height and depth were determined from averages 440 of the CPL data for this individual flight (due to operational constraints, this data set was fairly sparse). The albedo was constrained by spectro-radiometer observations. These observations are impacted by low clouds (not considered in the model) and thus surface albedo is likely biased high, although still in line with recent ice cloud RT modeling [Hong et al., 2016]. The size distribution of ice particles used in the RT model was parameterized as bimodal with peaks at 3 $\mu$m and 5 $\mu$m; the single scattering albedo of the ice particles was assumed to range from 0.982 to 0.999 in the 0.9-1.8 $\mu$m wavelength range. The choice





of these parameters was constrained by preliminary simulation, in line with the findings of other studies [Rosenfield et al., 1998, de Reus et al., 2009]. The average ice particle number was assumed to be the nominal value, but was varied in different RT model runs in order to retrieve SIWP. It should be noted that, for a fixed size distribution and cloud geometry, SIWP and number concentration are proportional. Based on Hawkeye observations of predominantly near-spherical particles, we use a Mie code to obtain the optical properties of the ice particles. Studies based on ATTREX data demonstrated how small particles increase

with decreasing temperature (higher altitude), and how irregularly shaped particles are evenly distributed within the particle size distribution [Woods et al., 2018]. In addition, efforts were made to reduce artifacts in the size distribution counting that might occur due to the fragmentation of large ice particles upon impact with the Hawkeye instrument during the measurements, thus making the counting of small ice crystals more reliable [Lawson, 2011, Thornberry et al., 2017].

## 4.2 IWP retrievals

We performed SIWP model calculations for three different ice concentrations (nominal value $n$, $n \times 0.5$, and $n \times 2$), defining geometry and RT parameters for each observation (Fig. 8, panel a). The modeled SIWP values bracket the observations and show the sensitivity of SIWP to ice water concentration. It is worth noting that the nominal ice concentration, associated with the average particle number density derived from the Hawkeye-FCDP instrument, already leads to a good general agreement with the observations (red line in Fig. 8, panel a), with the exception of a notable mismatch between roughly 5:40 am UTC

and 06:00 am. This difference is likely due to an actual IWP higher than the modeled IWP. The aircraft is descending at the beginning of this time interval, encountering more dense cirrus layers at lower altitudes that possess more than twice the averaged ice density assumed for the model. Another disagreement can be seen during the 7:26-7:40 am portion of the flight, where the model favors an ice density of about half of the nominal value. Overall, the comparison shows that observed SIWP and that calculated using the nominal ice concentration agree well for the remaining part of the observation time,

with differences mostly in the order of <1%-20% . Panels (c) and (d) in Fig. 8 show the flight altitude and azimuth angle, respectively, showing the rapid ascent and descent of the aircraft as it spirals around the same atmospheric region. Despite the swift changes in geometry, simulated and observed SIWP consistently agree, revealing the efficacy of VLIDORT-QS to simulate SIWP observation at the limb for different geometries.

To better understand the ability of this methodology to detect thin cirrus cloud, an IWC detection limit was determined from the

SIWP measurements. Time-series IWCs are retrieved by interpolating the measured SIWP onto the modelled SIWP. Because a fit between the data points is used to invert SIWP to IWC, a standard IWC error can be calculated from the SIWP error and the propagation of the fit uncertainty. Four different cases were chosen to represent the variability of the retrieved IWC from the measured SIWP. Retrieved IWCs for these four cases range from $5.4 \times 10^{-3}$ $\mathrm{g\,m^{-3}}$ to 0.02 $\mathrm{g\,m^{-3}}$, capturing a wide range of cirrus cloud conditions. The IWC error was determined by propagating the SIWP measurement error through the IWC retrieval

for those four cases. A mean error of $1.39 \times 10^{-4}$ $\mathrm{g\,m^{-3}}$ was found. Defining the IWC detection limit as three times this mean error (~99.7%), leads to an average IWC detection limit of $4.17 \times 10^{-4}$ $\mathrm{g\,m^{-3}}$.

**Figure 8.** *Comparison between the SIWP as measured by the mini-DOAS instrument and as simulated by the VLIDORT-QS model for three different ice number concentration values (panel (a)). Panel (b) shows the comparison of the ice water content (in g m$^{-3}$) retrieved from the mini-DOAS observations and as measured by the Hawkeye-FCDP and NOAA-Water instruments, for the portion of the flight considered. Flight altitude and azimuth angle are also shown (panel(c) and panel (d) respectively).*



### 4.3 IWP validation

TTL cirrus cloud micro-physical properties were widely sampled during the ATTREX 2014 campaign, at altitudes ranging from 14 km up to the cold point tropopause (17.5-18 km). In this region, cirrus clouds can extend vertically from one hundred
meters to a few kilometers, with cloud temperatures decreasing with altitude following the lapse rate in the TTL. These observations resulted in the determination of cirrus clouds IWC from water vapor TDL measurements (NOAA-Water instrument) and optical cloud probe (Hawkeye-FCDP instrument) observations [Thornberry et al., 2017]. The IWC retrieved from the observed mini-DOAS SIWP is then compared (Panel (b) in Fig. 8) to in situ measurements from these instruments for the same observational sequence. Due to spatial and temporal averaging of the mini-DOAS observations, a moving average of 4 minutes
has been applied to the Hawkeye-FCDP/NOAA-Water data. This smooths some of the high IWC variability observed by these high frequency in situ instruments.

For the portion of the flight considered here, LIDAR images from the CPL instrument showed consistent cirrus structures mostly between 15 km and 16.5 km, with a mixed-cloud layer beneath (around 14-15 km). This can be clearly seen in Fig. 8, where the Hawkeye-FCDP/NOAA-Water IWC values peak when the aircraft passes through the top and bottom of the cloud
during ascending and descending flight stages (panel (c) in Figure 8). During the 15-16.5 km cloud transects around 6:15 am, 6:55 am, and 7:15 am, the in situ and mini-DOAS IWC data show good agreement in the core of the cloud. The comparison is less favorable for the measurements in the lower cloud deck, where the mini-DOAS is smaller (5:45 am) or larger (7:40 am). During the other periods, the mini-DOAS IWC is generally higher than that measured by the Hawkeye-FCDP and NOAA-Water instruments.

The IWC retrieved from the mini-DOAS observations seems to be more constant along the flight and less able of capturing the small scale variability of the ice. The reason for this difference is explained by the spatial averaging of the mini-DOAS instrument along the LOS. Due to the high altitude and limb viewing geometry, observations are averaged over a very long path in the atmosphere that can reach up to 300 km ahead of the aircraft. The mini-DOAS will thus observe clouds even if the GH is flying in clear air that leads to near zero or very low in situ Hawkeye-FCDP/NOAA-Water IWC.

It is well established that cirrus cloud micro-physical properties are the result of a complex interaction of different parameters [Jensen et al., 2010], and that the variability in these parameters is the source of significant heterogeneity in the total IWC on both horizontal and vertical scales. This diversity can be measured by the change in IWC (or a related parameter) in the sampling step (at 1 second sampling rate, a high frequency instrument such as the NOAA-Water covers roughly 160 meters horizontal distance and 35 meters vertical for each data point). These in-situ instruments are thus more sensitive to the local
environment conditions at the time of acquisition, while the mini-DOAS gives a more averaged view of the presence of cirrus clouds.

Measurement comparisons between the different instruments, therefore, have to be interpreted with much care as the instruments probe different air masses. Nevertheless, the comparisons show that, considering these limitations, our retrieved IWC from the mini-DOAS are comparable to those found inside cirrus clouds in the general observational area.




## 4.4 Retrieval uncertainty

In order to evaluate the uncertainties introduced by a change of the RT input parameters to the retrieved IWP for the flight segment shown in Fig. 8, we performed sensitivity tests for different ice cloud heights and particle size distributions. We exclude situations where the aircraft was above the clouds, as limb observations of SIWP are little sensitive under these

conditions. Limb observations are also challenging when the aircraft is in clear air below cirrus clouds, although retrievals are still possible. These overhead cloud situations can be observed more accurately by looking slightly upwards from the limb, something we did not explore in this study. Consequently, we are not considering retrieval uncertainties for these cases in this section. For the remaining cases, LIDAR data provided information on the altitude and the extent of the ice clouds for the portion of the flight considered, allowing the model to find the best fit between measured and modeled SIWP, with a nominal

cloud located at 14-18 km.

To evaluate the uncertainties in the retrieved IWP associated with different cloud heights, we varied the geometrical extent of the cloud by three fixed quantities $\Delta h$ (0.5, 1, and 2 km), subtracting $\Delta h$ from the cloud top only, the cloud bottom only, and the cloud top and bottom simultaneously. The sensitivity tests revealed a significant dependence on the SAZ, making it difficult to assign a single value on IWP uncertainties imposed by a poorly known cloud height. We therefore report uncertainties, i.e.,

deviations from the base case, for the extreme values of SAZ for each case in Table 5.

When the vertical extent of the cloud in the retrieval is reduced by 12.5% ($\Delta h = 0.5$ km) from the top or bottom, IWP uncertainties are on the order of 5%-10% (Table 5). When the same quantity $\Delta h$ is applied to both top and bottom (cloud is now reduced by 25%), modeled IWP differ from the best retrieval by 10% - 15%. Uncertainties increase when a reduction of $\Delta h = 1$ km is applied to the nominal cloud. In this case, a reduction of 25% from the top yields 12-15% uncertainty depending

on azimuth, while a reduction of the cloud extent from the bottom produces 8% - 20% uncertainties. When $\Delta h$ is applied simultaneously to both top and bottom (cloud reduced by 50% from the nominal value), IWP uncertainties roughly double to 20% - 35%. For a further reduction of the cloud from the top ($\Delta h = 2$ km, 50% extent reduction), uncertainties increase to 30% - 40%, while the same reduction from the bottom leads to uncertainties in the order of 20% and 60%.

Overall, IWP uncertainties increase when the actual cloud geometrical extent differs from that for the nominal cloud used in

the retrieval. This behavior appears to be relatively linear for a smaller cloud extent decrease (25% the nominal height) for the top-only or bottom-only cases. When the reduction of the cloud height is more severe (50%), the uncertainties of the modeled IWP become larger and depend more strongly on SAZ. It should be noted that, as IWP is the IWC integrated over the vertical extent of the cloud, the IWP uncertainties due to variations in cloud extent are solely due to the RT simulations. In order to reduce these uncertainties, additional methodologies, for example based on measured radiances or $O_2$ absorptions, are needed

to constrain cloud extent when LIDAR data are unavailable.

The sensitivity of IWP retrievals to ice particle habit is beyond the scope of this study. However, we have evaluated the impact of uncertainty in the ice particle size distribution. The base IWP retrieval used a bimodal distribution with peaks at 3 $\mu$m and 5 $\mu$m. To test the influence of an uncertain size distribution, we increased the diameter of both peaks of the bimodal distribution by 1 $\mu$m and performed a retrieval of our data. The IWP changed by no more than 6%, showing a low sensitivity to a change





| Δh | Cloud Top | | Cloud Bottom | | Cloud Top/Bottom | |
|---|---|---|---|---|---|---|
| | $\alpha = 90°$ | $\alpha = 300°$ | $\alpha = 20°$ | $\alpha = 220°$ | $\alpha = 45°$ | $\alpha = 90°$ |
| -0.5 km | 7% | 5% | 5% | 10% | 10% | 15% |
| -1.0 km | 15% | 12% | 8% | 24% | 20% | 35% |
| -2.0 km | 40% | 30% | 20% | 60% | - | |

**Table 5.** *IWP uncertainties for three different cloud extent reduction Δh, for cloud top only, cloud bottom only, and both cloud top and bottom, and for different azimuth angles $\alpha$.*

in the particle size and distribution. This result is encouraging, as obtaining particle size distributions for ice is difficult from remote sensing observations alone [Nakajima and King, 1990; McFarquhar and Heymsfield, 1997; BARAN and HAVEMANN, 2004].

## 5 Conclusions

This study explores the measurement of cirrus cloud ice water path using limb-viewing optical remote sensing of ice absorption

in the near infrared wavelength region. Our approach has not been reported previously in the literature, and this has required the development of new spectral retrieval methods as well as an enhanced RT model.

Spectral retrievals of the ATTREX mini-DOAS near-IR limb observations have clearly identified ice absorption in cirrus clouds, through the retrieval of the slant ice water path, i.e., the integrated ice absorption along the complex photon paths through the atmosphere. In addition, $O_2$ and $CO_2$ absorption can successfully be retrieved in this wavelength region, providing additional

information on the atmospheric RT.

We developed and applied a new extended version of the widely-used VLIDORT model, called VLIDORT-QuasiSpherical, which has the ability to simulate limb observations accurately. VLIDORT-QS is fully linearized and has the ability to generate simultaneous fields of Stokes 4-vectors and analytically-derived Jacobians with respect to any atmospheric quantity, making it eminently suitable for forward-model simulations and sensitivity and retrieval studies.

Using these new tools, we performed a comprehensive sensitivity test of ice absorption in cirrus cloud for different parametrizations, and subsequently developed a conceptual model of this measurement approach. This fundamental understanding of airborne near-IR limb observations is crucial for future efforts to explore this approach to study cirrus clouds from aircraft or satellites. Our conceptual model identified two general regimes:

1. A linear regime for thin and moderately thick clouds in which the ice absorption is proportional to the IWC

2. A regime for thick cirrus clouds in which the ice absorption saturates and becomes independent of IWC

Oxygen absorption does not contribute much information to the first regime, but can be useful to study optical properties of ice particles in the second case. This was demonstrated in our paper, but not further explored.





A feasibility retrieval of IWC from the limb mini-DOAS observations was also developed. This interpolation-based approach uses a small number of RT calculations for the viewing geometry of each observation to determine the sensitivity of SIWP to

IWC. The retrieved IWCs also agree with those measured in situ by both the Hawkeye-FCDP and NOAA-Water instruments, although, due to the nature of its observation geometry, we found that the mini-DOAS IWC provides information on a much larger spatial scale and is thus less able to capture the high IWC variability inside cirrus clouds. Finally, this approach showed its lower detection sensitivity for IWC in limb geometry and the potential for detection of cirrus at low ice water concentration. Our study has demonstrated the value of near-IR absorption measurements to study cirrus clouds. As near-IR spectrometers

are small and their technology is mature, our work points to a new way to measure cirrus cloud properties remotely.

*Competing interests.*  Jochen Stutz is a member of the editorial board of Atmospheric Measurement Techniques.

*Acknowledgements.*  This study was funded by the NASA Upper Atmosphere Research Program (NASA ATTREX Grant numbers NNX10AO80A for the mini-DOAS measurements). Additional support for the mini-DOAS measurements came through the Deutsche Forschungsgemein-schaft, DFG (through grants PF-384 5-1/2, PF384 7-1/2 PF384 9-1/2, and PF384 12-1), and the EU project SHIVA (FP7-ENV-2007-1-226224).


*Author contributions.*  **S.F. Colosimo**: Methodology, Validation, Visualization, Writing - Original Draft, Review & Editing. **N. Brockway**: Methodology, Writing, Review & Editing. **V. Natraj**: Review & Editing. **R. Spurr**: Software, Validation, Review & Editing. **K. Pfeilsticker**: Review & Editing. **L. Scalone**: Review & Editing. **M. Spolaor**: Review & Editing. **S. Woods**: Review & Editing. **J. Stutz**: Supervision, Methodology, Writing, Review & Editing

585    .



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
