# Peer review of "Investigation of cirrus clouds properties in the Tropical Tropopause Layer using high-altitude limb scanning near-IR spectroscopy during the NASA-ATTREX Experiment"

_Atmospheric Measurement Techniques, 2023_

## Referee Comment (RC1)

Review of "Investigation of cirrus clouds properties in the Tropical Tropopause Layer using high-altitude limb scanning near-IR spectroscopy during the NASA-ATTREX Experiment", by Colosimo and coauthors, AMT-2023-85

The focus of this article is on the mini-DOAS (Mini Differential Optical Absorption Spectroscopy) instrument, and its operation on the NASA Global Hawk aircraft during the Airborne Tropical TRopopause Experiment. The instrument provides limb scanning observations in the near-IR, facilitating the identification of ice and liquid water. To use the instrument's capability, radiative transfer code was developed for this study. Comparison of the ice water path and ice water content retrieved from the instrument agreed reasonably well with the observations from the SPEC FCDP and the NOAA water vapor instrument.

I will focus on the observations with the particle probes (Hawkeye) and the comparison with the observations as this is my area of expertise. I'll let others comment on the radiative transfer calculations and instrument design and capability.

Main comments

Line 68. Hawkeye is subject to considerable ice particle shattering. You briefly comment on this later in the article but I think it should be here. Also, I'm not convinced that the shattering removal techniques for the small particles effectively reduces or eliminates shattering. We have quite a lot of good data to show this.

183-185, 487-489. Use LIDAR to get the IWC and IWP. You can use LIDAR extinction data and a relationship between extinction and ice water content to also get SIWC and ice water path and to compare with the mini-DOAS instrument. See Heymsfield et al. (2014). Heymsfield, A., D. Winker, M. Avery, M. Vaughan, G. Diskin, M. Deng, V. Mitev, and R. Matthey, 2014: Relationships between Ice Water Content and Volume Extinction Coefficient from In Situ Observations for Temperatures from 0° to −86°C: Implications for Spaceborne Lidar Retrievals. *J. Appl. Meteor. Climatol.*, **53**, 479–505.

266. Are the sizes of the particles imaged by the CPI (their maximum dimensions) consistent with those sampled by the FCDP, because it's possible that particles >50 microns were present. In Woods et al. (2018), CPI images of particles as large as several hundred microns are shown (their Figure 5).

272, 448. Particle habit. I don't necessarily agree that the particles are quasi-spherical. Are the images from the CPI consistent with spherical particles-ice density of 0.91 g/cm^3? You can clearly see this in Fig. 5 of Woods et al. (2018). Please comment on this. Also, see: Heymsfield, 1986: Ice particles observed in a cirriform cloud at −83 °C and implications for polar stratospheric clouds. *Journal of Atmospheric Science*, 43(8), 851–855.

The long averaging times of the mini-DOAS instrument are somewhat problematic.

Minor Comments

What were the temperatures sampled?

Line 43-45. CALIPSO/CALIOP can readily detect thin TTL cirrus.

70. Here you should mention how the IWC was derived from the Hawkeye instrument. You use the Hawkeye IWCs in Figure 8.

123. Can LIDAR extinction be used to derive absorption cross-sections of ice? Can the two be related? You do have the LIDAR data.

Eq. (5) and line 164. Is this the density of solid ice? Is $M_{ice}$ the ice water content (IWC)? That's what it should be.

300. I like the sensitivity tests you did, varying the concentration and examining the results.

451-453. This should be inserted earlier, where Hawkeye is discussed.

457-460. Does the discussion here relate to Figure 8b? You do discuss Figure 8b later but it probably should be here. I really don't see the good agreement. Also, by "observations", do you mean the NOAA instrument?

---

## Referee Comment (RC2)

A review on 'Investigation of cirrus clouds properties in the Tropical Tropopause Layer using high-altitude limb scanning near-IR spectroscopy during the NASA-ATTREX Experiment'

by Santo Fedele Colosimo et al.

submitted to Atmospheric Measurement Techniques

This study shows a new method to derive IWC from ice water absorption derived from NASA ATTREX MiniDOAS limb scanning near-IR observations. A case study is used to interpret the retrievals, validations, and uncertainties. Sensitivity tests were performed using a radiative transfer model, and a conceptual model was derived to understand observations.

This paper is well-organized and well-written. However, I still have some concerns.

1. This study uses one case to show the performance of IWC retrievals from MiniDOAS, validated with in-situ measurements from Hawkeye FCDP and NOAA water instruments. Validation shows good agreement at some times but not that good at another time. The authors discussed possible factors that contribute to the uncertainty. However, with only one case, it is not that convincing how good the method is to derive IWC. I would suggest adding other cases (as complementary materials).

2. Please check the equations 10 -13.

3. Throughout the entire paper, the authors may state the same concept in various places but use different terminology, which is confusing. These include concentration, number concentration, mass concentration, particle density, also IWC and IWP. Please keep the terms consistent, correct, and easy to understand for readers.

4. An important conclusion in this paper is the high sensitivity of the method to detect and derive information for thin cirrus clouds. However, there is no comparison or discussion with other methods to prove the method's sensitivity.

Specific comments:

Line 2: ' High altitude aircraft offer', aircraft→ aircrafts

Fig 1. Please define optical density, and explain its usage in this study. Y titles of fig.1 'OD' is not expanded. Readers may be misled by optical depth. Please clarify it.

Line 160: 'IWP is defined as the vertical integral of ice mass concentration through a cloud'. Why not just use ice water content instead of ice mass concentration? It seems easy to confuse it with concentration, number concentration, and number density in this paper.

Line 163: 'Mice' why not just IWC? It will directly relate to the retrieval of IWC from SIWP later in the paper.

Line 187: add ',' between 'angle One'.

Line 190: what does 'SIWP tracks the O2 SCD' mean?

Line 192: 'This is somewhat counter-intuitive, considering ....' Why that happens?

Line 267: 'FCDP measurements yielded the number density of the particles', please define number density.

Line 272-274: 'using the ice volume related to the averaged particle number concentration n', are you sure you mean particle number concentration, not the mass concentration as mentioned above?

'This resulted in a nominal ice concentration of $N_{ice}$ = 1.22 x 10$^{-3}$ g m-3' Obviously, the unit here refers to mass not number concentration. I think it has the same meaning as $M_{ice}$ in Equation (5). To get grid of confusion, please keep the symbol consistent. Again, I would suggest using IWC instead of mass concentration.

'n is referred to as the nominal case', do you mean n = $N_{ice}$? Then n is mass concentration not mass concentration?

Equations 6 and 7: Could you explain more why it needs two wavelengths to retrieve $\tau_{ice}$, but only one wavelength to retrieve $\tau_{O2}$?

Table 3: it is confusing to read.

Line 301-306: my understanding here ice concentration refers to ice mass concentration. n = 1.22*10-3 g/m3.

Figure 3: for y titles, replace ice absorbance with SIWP, and oxygen absorbance with SCD, if I understand correctly. Please add units for both x and y axis. For x axis, what does ice nominal x n mean? I think n represents ice nominal case.

Equations (10) and (11): the radiation being reflected by surface or atmosphere below the aircraft may transmitted through clouds. Why extinction of clouds is ignored? Which is to say, instead of $I_s(\lambda)R(\lambda)$ , it may be more realistic using $I_s(\lambda)R(\lambda)T(\lambda)$, T is the transmissive.

Line 379 ' $N_{ice}$ identifies the ice particle number density' I think it is called nominal ice concentration in Line 273, which should be mass concentration, not particle number density. Again, please keep the terms and symbols consistent if you are meaning the same thing.

Equation 13: I think $\tau = \ln\left[\frac{I_{obs}}{I_S}\right]$, based on Equation 7. Then Equation 13 misses a negative sign.

Table 4: size distribution 'bimodal 3-5 um', change as 'bimodal peak 3-5 um' might be clearer.

Line 446: 'the average ice particle number' is it the ice particle mass concentration as mentioned in Line 273? Note that particle number and particle mass are two different terms.

Line 447- 448: 'SIWP and number concentration are proportional', be sure if you mean the number concentration or mass concentration.

Line 460: 'This difference is likely due to an actual IWP higher than the modeled IWP'. This is confusing. SIWP? Ice concentrations?

Line 463: ' the model favors an ice density', is it the observation favors an ice densityof about half of the nominal value? I think the nominal values is used in the model.

Line 470-480: Please explain more specifically how to get IWC from SIWP? Is there any equations or references to point the readers to a better understanding of the methods?

Line 472: 'Four different cases were chosen...' please add details (date, location, flight altitude etc.) about these four cases.

Line 473: 'retrieved IWCs for these four cases range from 5.4x10^-3 g/m3...', how well is this limit compared to lidar?

Section 4.3: the whole section discusses IWC rather than IWP.

Figure 8, panel b, the unit of IWC 'gr m-3' → g m-3 to keep it consistent in the paper.

Line 512: Fig. 8 shows SIWP not 'retrieved IWP'. Do you mean IWC?

Lines 521-525: It is confusing how to get IWPs and their uncertainties in this section. What is the best retrieval? How do you get the percentage uncertainty?

Line 538: 'as IWP is the IWC intenerated over the vertical extent' move the definition of IWP before the discussion of IWP uncertainty, and explain a little bit on why discuss IWP not IWC.

Line 573: 'lower detection sensitivity for IWC in limb geometry'. Do you mean higher detection sensitivity? Compare to what other approaches to demonstrate a better sensitivity of your approach?

---

## Author Comment (AC1)

**Authors' response to comments from Anonymous Referee #1**

**Referee #1**
Review of "Investigation of cirrus clouds properties in the Tropical Tropopause Layer using high-altitude limb scanning near-IR spectroscopy during the NASA-ATTREX Experiment", by Colosimo and coauthors, AMT-2023-85

The focus of this article is on the mini-DOAS (Mini Differential Optical Absorption Spectroscopy) instrument, and its operation on the NASA Global Hawk aircraft during the Airborne Tropical TRopopause Experiment. The instrument provides limb scanning observations in the near-IR, facilitating the identification of ice and liquid water. To use the instrument's capability, radiative transfer code was developed for this study. Comparison of the ice water path and ice water content retrieved from the instrument agreed reasonably well with the observations from the SPEC FCDP and the NOAA water vapor instrument.

I will focus on the observations with the particle probes (Hawkeye) and the comparison with the observations as this is my area of expertise. I'll let others comment on the radiative transfer calculations and instrument design and capability.

We would like to thank the referee #1 for providing valuable and constructive comments, as well as suggestions to improve the manuscript. Responses to these specific comments are provided below.

Main comments

**Line 68**. Hawk-eye is subject to considerable ice particle shattering. You briefly comment on this later in the article, but I think it should be here. Also, I'm not convinced that the shattering removal techniques for the small particles effectively reduces or eliminates shattering. We have quite a lot of good data to show this.

The Hawkeye instrument is subject to shattering under certain observational conditions, particularly in the presence of moderate to high concentrations of large ice particles, but most of the ATTREX flights did not encounter such conditions and had low evidence of shattering. Shattering has been mentioned later in the manuscript (line 451) to note our awareness of this issues, and the fact that corrections of this effect have been made (according to the cited reference) to improve data reliability. Further investigation of minor shattering artefacts are beyond the scope of the current work.

The aim of this section is to provide a list of the instruments involved in the ATTREX project that are relevant to this study, leaving the description of potential issues related to

the use/application of their data to parts of the manuscript where the data are discussed. Nevertheless, we agree that mention of the shattering effect should be moved to this paragraph as a discussion of the operational limitation of the instrument. Consequently, we have moved the paragraph forward to Section 2, following the reviewer's suggestion.

**Line 183-185, 487-489**. Use LIDAR to get the IWC and IWP. You can use LIDAR extinction data and a relationship between extinction and ice water content to also get SIWC and ice water path and to compare with the mini-DOAS instrument. See Heymsfield et al. (2014). Heymsfield, A., D. Winker, M. Avery, M. Vaughan, G. Diskin, M. Deng, V. Mitev, and R. Matthey, 2014: Relationships between Ice Water Content and Volume Extinction Coefficient from In Situ Observations for Temperatures from 0° to −86°C: Implications for Spaceborne Lidar Retrievals. J. Appl. Meteor. Climatol., 53, 479–505.

We agree that the comparison of IWC and IWP LIDAR data to values obtained from the mini-DOAS would have provided further validation to our study. However, a direct comparison of the two remote sensing methods is quite challenging due to the very different viewing strategies, i.e., nadir vs limb as well as the averaging volumes (or kernels). In addition, the use of LIDAR as a tool to measure IWP also requires radiative transfer models, thus adding more uncertainty. We have therefore considered that the comparison of our results with two direct in-situ IWC measurements was sufficient to validate our methodology.

**Line 266**. Are the sizes of the particles imaged by the CPI (their maximum dimensions) consistent with those sampled by the FCDP, because it's possible that particles >50 microns were present. In Woods et al. (2018), CPI images of particles as large as several hundred microns are shown (their Figure 5).

We are aware of the limitations of the use of the Hawkeye-FCDP data. This instrument only measures particles between about 1–50 microns, relying on the Hawkeye-2D-S to cover a larger size range (10 microns to a few mm). We performed sensitivity studies and found that inclusion of larger particles in the radiative transfer models yielded unrealistic results. We therefore believe that these particles may not have been present at large concentrations in the air volume observed by our instrument. We agree that a more accurate description of the particle size distribution would have been beneficial for the presented methodology, but we consider that constraining the size range of the in-situ microphysical data for more direct comparison with the SIWP retrievals to be acceptable, given the scope of the paper.

**Line 272, 448**. Particle habit. I don't necessarily agree that the particles are quasi-spherical. Are the images from the CPI consistent with spherical particles ice density of 0.91 g/cm^3? You can clearly see this in Fig. 5 of Woods et al. (2018). Please comment

on this. Also, see: Heymsfield, 1986: Ice particles observed in a cirriform cloud at −83 °C and implications for polar stratospheric clouds. Journal of Atmospheric Science, 43(8), 851–855.

CPI images for that portion of the flight considered in this study mostly show spherical and quasi/spherical shaped particles, with virtually none of the other detectable habits present. The Heymsfield reference suggests they observed trigonal and columnar habits, but the observed ice particle population for ATTREX included very few trigonals, and few columns in comparison to quasi-spheroids. Also, in Woods et al. (2018), the authors often refer to spherical or quasi-spherical properties of the ice particles in their paper, in particular treating these properties as an a priori assumption for some methodologies. They have found that assuming spherical, which allows use of the FCDP data, is a good approximation within the larger uncertainty bounds of applying the FCDP measurements here. Decreasing the uncertainties inherent in the FCDP measurements is beyond the scope of the current work, but the authors do acknowledge the relevance of the comments and will consider such issues in future in-situ focused cloud and instrumentation studies.

From Woods et al. (2018) - Section 2: Measurements

- Page 6056: "*From the concentration and sizing, particle area and mass over this size range are estimated assuming spherical ice particles since the exact shape of the particles is not known.*"

- Page 6057: "*…the portion of the size distribution smaller than the cutoff is from FCDP observations (assuming circular area and spherical mass)*"

The validity of the assumption of spherical particles (and thus the use of a Mie code for calculating the optical properties) was also confirmed through the good agreement between the modeled and measured SIWP considered in this study.

The long averaging times of the mini-DOAS instrument are somewhat problematic.

The mini-DOAS data temporal resolution (and large averaging volume) can represent a challenge for a direct comparison with instruments at higher data frequency. In order to validate the retrieved mini-DOAS IWC, temporal smoothing was applied to data from the in-situ instruments. We are aware of the limitation of this comparison (as commented on in Section 4.3). However, the comparison still holds, showing the feasibility to retrieve comparable IWC using near-IR limb measurements.

Minor Comments

What were the temperatures sampled?

According to the in-situ data for that portion of the flight considered in this study (14–18km flight altitude), temperatures were recorded in the 185–210K range. The sentence: "*…where temperatures were recorded in the 185–210 K range.*" has been added at the end of the existing paragraph in lines 483–484:

"*TTL cirrus cloud micro-physical properties were widely sampled during the ATTREX 2014 campaign, at altitudes ranging from 14 km up to the cold point tropopause (17.5–18 km), where temperatures were recorded in the 185–210 K range.*"

**Line 43-45**. CALIPSO/CALIOP can readily detect thin TTL cirrus.

We rephrased the paragraph in line 43-45 as follows:
"*While cirrus clouds occur frequently in the TTL [Wylie and Menzel, 1999], they are often optically thin and thus difficult to observe. Spaceborne experiments have provided important information about the properties of these type of clouds [Winker, 2009; Winker 2010].*"

**Line 70**. Here you should mention how the IWC was derived from the Hawkeye instrument. You use the Hawkeye IWCs in Figure 8.

As explained in the reply to the comment for Line 68, the aim of this section is just to list the instruments that were part of the ATTREX payload, generating data related to the analysis carried out in this study. Hawkeye data provided the size distribution of the particles, which has been averaged for the portion of the flight considered, in order to evaluate the total volume. Knowing the density of ice, we evaluated the nominal ice water content (renamed $IWC_0$). The paragraph has been rephrased later in the text (lines 267–269) for a clearer explanation.

**Line 123**. Can LIDAR extinction be used to derive absorption cross-sections of ice? Can the two be related? You do have the LIDAR data.

LIDAR cannot be used to derive ice absorption cross section. The Cloud Physics Lidar instrument provides the extinction-to-backscatter parameter (among other output products) and not the absorption cross section. CPL also operates at three specific wavelengths: 1064, 532, and 355 nm, (McGill, 2002), that do not overlap with any strong ice absorption features between about 1450 nm to 1550 nm, used in the present study.

**Eq. (5) and line 164**. Is this the density of solid ice? Is Mice the ice water content (IWC)? That's what it should be.

Equation (5) is only a theoretical explanation of the SIWP, and the formula is not directly used for calculations in the analysis. SIWP is expressed in length units (um) so M represents the mass concentration of ice (or IWC), and $\rho_{ice}$ the ice density. We agree that $M_{ice}$ is equivalent to IWC, and this could be somewhat misleading to the reader. The term $M_{ice}$ has been replaced with IWC in the text.

**Line 300**. I like the sensitivity tests you did, varying the concentration and examining the results.

We thank the reviewer for this kind comment.

**Line 451-453**. This should be inserted earlier, where Hawkeye is discussed.

As mentioned in the Line 68 comment, this sentence has been moved to Section 2.

**Line 457-460**. Does the discussion here relate to Figure 8b? You do discuss Figure 8b later, but it probably should be here. I really don't see the good agreement. Also, by "observations", do you mean the NOAA instrument?

We thank the reviewer for this comment. Answers to these two questions are next.

- No, this entire paragraph refers to Figure 8, panel (a). It was only to highlight how the nominal case represents the "best fit" when compared with the half (x0.5) and double (x2) cases.
- No, the term "observations" still refers to panel (a) in Fig.8, where SIWP is calculated from mini-DOAS observations.

---

## Author Comment (AC2)

**Authors' response to comments from Anonymous Referee #2**

**Referee #2**
A review on 'Investigation of cirrus clouds properties in the Tropical Tropopause Layer using high-altitude limb scanning near-IR spectroscopy during the NASA-ATTREX Experiment' by Santo Fedele Colosimo et al. submitted to Atmospheric Measurement Techniques.

We would like to thank the anonymous referee #2 for agreeing to review the manuscript, providing valuable and constructive comments and suggestions for improvement. Responses to the main and specific comments are provided below.

This study shows a new method to derive IWC from ice water absorption derived from NASA ATTREX MiniDOAS limb scanning near-IR observations. A case study is used to interpret the retrievals, validations, and uncertainties. Sensitivity tests were performed using a radiative transfer model, and a conceptual model was derived to understand observations.
This paper is well-organized and well-written. However, I still have some concerns.

1.  This study uses one case to show the performance of IWC retrievals from MiniDOAS, validated with in-situ measurements from Hawkeye FCDP and NOAA water instruments. Validation shows good agreement at some times but not that good at another time. The authors discussed possible factors that contribute to the uncertainty. However, with only one case, it is not that convincing how good the method is to derive IWC. I would suggest adding other cases (as complementary materials).

We thank the reviewer for bringing up this point. The main purpose of the paper was to develop and investigate the feasibility of using ice absorption to retrieve IWC within reasonable uncertainties, and to demonstrate the use of DOAS near-IR limb measurements as an additional tool to add to existing techniques for IWC retrieval in cirrus clouds. While we acknowledge that the validation would have benefited from a more extensive analysis of the entire data set, we decided to verify the methodology for a specific data segment during the ATTREX experiment, where the variability of the environmental radiative conditions of the measurements, such as solar position and altitude, allowed for a more thorough testing of the VLIDORT-QS (QuasiSpherical) RT model. As explained in Section 2.4, this RT model has been developed specifically for this project and the choice of that particular portion of the flight (Section 2.3) was driven by a continuous change in all of the RT parameters, which makes the validation of the RT code and the methodology more valuable within the scope of this study.

2. Please check the equations 10 -13.

We do not understand what specifically needs to be checked in equations 10 to 13. We do not see any errors in the equations as written.

3. Throughout the entire paper, the authors may state the same concept in various places but use different terminology, which is confusing. These include concentration, number concentration, mass concentration, particle density, also IWC and IWP. Please keep the terms consistent, correct, and easy to understand for readers.

We thank the reviewer for this helpful suggestion. We have carefully revised the manuscript to make the use of variables consistent.

4. An important conclusion in this paper is the high sensitivity of the method to detect and derive information for thin cirrus clouds. However, there is no comparison or discussion with other methods to prove the method's sensitivity.

The idea behind this study was to test the feasibility of detecting IWC in cirrus clouds from high altitude DOAS near-IR limb measurements, and a formal intercomparison of our methodology with other methods was not originally planned as part of the study. We did compare the retrieved miniDOAS IWC with other in situ IWC measurements taken on board the Global Hawk. The similarity of the results is considered enough given the scope of the paper. Furthermore, airborne IWC measurements in stratospheric cirrus clouds are limited, making a formal intercomparison between different techniques challenging. Notwithstanding, we do understand the reviewer's point, and we have added the following sentence at line 479:

"*It is worth noting that the NOAA instrument has an IWC detection limit of 2-3 g $m^{-3}$ for ideal conditions [Thornberry, 2015; Thornberry, 2017]. While our IWC detection limit is not as good as that for the NOAA instrument, our approach is not affected by sampling artefacts (i.e., ice shattering), and it is sufficiently different and sensitive to provide new insights into cirrus clouds.*"

Specific comments:

**Line 2**: 'High altitude aircraft offer', aircraft→ aircrafts

The term aircraft can be equally used both in singular and plural form (see https://www.britannica.com/dictionary/aircraft).

**Fig 1**. Please define optical density and explain its usage in this study. Y titles of fig.1 'OD' is not expanded. Readers may be misled by optical depth. Please clarify it.

We thank the reviewer for spotting this typo. As explained in Section 2, the DOAS analysis implements the Beer's Law (in the form of logarithm of the ratios of near-limb and direct-sun spectra), where the attenuation of the light through the medium is defined by the optical depth. In Fig.1, OD is an abbreviation for optical depth and not optical density, as erroneously reported in the caption of the figure. The caption has been corrected accordingly.

**Line 160**: 'IWP is defined as the vertical integral of ice mass concentration through a cloud'. Why not just use ice water content instead of ice mass concentration? It seems easy to confuse it with concentration, number concentration, and number density in this paper.

We agree with the reviewer about the confusion that the term mass concentration can generate. The term ice mass concentration has been replaced with IWC.

**Line 163**: 'Mice' why not just IWC? It will directly relate to the retrieval of IWC from SIWP later in the paper.

As explained in the previous comment, we agree that $M_{ice}$ is equivalent to IWC and somewhat misleading to the reader. $M_{ice}$ has been replaced with IWC.

**Line 187**: add ',' between 'angle One'.

The missing dot has been added to the text.

**Line 190**: what does 'SIWP tracks the O2 SCD' mean?

It means that SIWP behaves similarly to the O2 SCD.

**Line 192**: 'This is somewhat counter-intuitive, considering ….' Why that happens?

We do not have a conclusive answer for this, but we have provided a balance between two different mechanisms as a possible explanation. This is mentioned directly after the above sentence.

**Line 267**: 'FCDP measurements yielded the number density of the particles', please define number density.

Number density here refers to the size distribution of the particles, i.e., number of particles per specific radius. The sentence has been rephrased as:
" *FCDP measurements yielded the number concentration as a function of particle size.* "

**Line 272-274**: 'using the ice volume related to the averaged particle number concentration n', are you sure you mean particle number concentration, not the mass concentration as mentioned above?

The term refers to number concentration as defined in the answer to the previous comment. Hawkeye data provided the size distribution of the particles, which has been averaged for the portion of the flight considered, in order to evaluate the total volume. Knowing the density of ice, we evaluated the nominal ice water content $IWC_0$.

'This resulted in a nominal ice concentration of $Nice$ = 1.22 x 10-3 g m-3' Obviously, the unit here refers to mass not number concentration. I think it has the same meaning as $Mice$ in Equation (5). To get grid of confusion, please keep the symbol consistent. Again, I would suggest using IWC instead of mass concentration.

The reviewer is correct. It does have the same physical meaning. The term nominal ice mass concentration $N_{ice}$ has been replaced with $IWC_0$, representing the nominal ice water content.

'n is referred to as the nominal case', do you mean n = $Nice$? Then n is mass concentration not mass concentration?

Yes, *n* refers to the mass concentration. In line with keeping the symbols consistent, the term *n* has been deleted to avoid confusion, and the paragraph from lines 267–271 has been changed to:

"*We therefore calculated an averaged ice particle concentration over the period of interest for this study and used this to calculate the relative total volume. Knowing the density of ice $\rho_{ice}$ = 0.91 × 10^{-6} g m−3, this resulted in an ice water content of $IWC_0$ = 1.22 × 10^{-3} g $m^{-3}$. If not otherwise specified, $IWC_0$ is referred to as the nominal case in the rest of the paper. This value represents the baseline, and the different ice water content values used in this study are expressed as multiplicative factors with respect to the nominal value.*"

**Equations 6 and 7**: Could you explain more why it needs two wavelengths to retrieve $\tau ice$, but only one wavelength to retrieve $\tau O2$?

According to Beer law's, trace gas optical depth can be retrieved knowing intensity values with and without the gas of interest. This can be easily done with two separate runs of the RT code. For ice, this is not possible, because of the large difference in magnitude of the intensity with and without ice. Beer's law can however still be applied, using two wavelengths close enough to each other (1548.1 nm and 1550.3 nm in our study) to assume a similar ice absorption but far from the main ice absorption band. These two approaches are described by Eq. (7) and Eq. (6), respectively.

**Table 3**: it is confusing to read.

We thank the reviewer for the comment. For a better understanding of the table content, we have rephrased the caption as:

"*The table lists the parameters used for the three different sensitivity tests (with respect to geometry, surface emissivity, and cloud altitude) performed. For example: geometry test has a fixed surface albedo (0.05) and cloud extension (14–18km), and only solar and azimuth angles are varied, according to the value in the corresponding cell. Flight altitude is set to 16.5 km for all tests.*"

**Line 301-306**: my understanding here ice concentration refers to ice mass concentration. n = 1.22*10-3 g/m3.

The reviewer is correct. As already explained in the response to Line 160 comment (and following), the nominal value was replaced with $IWC_0$ in line 268.

**Figure 3**: for y titles, replace ice absorbance with SIWP, and oxygen absorbance with SCD, if I understand correctly. Please add units for both x and y axis. For x axis, what does ice nominal x n mean? I think n represents ice nominal case.

We thank the reviewer for this comment. The Y-axis for both the ice and $O_2$ tests refer to absorbance (dimensionless), as shown through the explanations in Eq.(6) and Eq.(7), and not SIWP or SCD. The purpose of this sensitivity test is only to show how ice and $O_2$ absorbance behave when the ice mass concentration is varied. We do state in line 302 that ice absorbance and SIWP are proportional, implying that the behavior of ice absorbance is proportionally related to the SIWP.

We agree with the reviewer's comment on the X-axis. The X-label has been changed to "*x $IWC_0$ [g $m^{-3}$]*" and the sentence "*The X-axis values represent the different ice mass concentrations, expressed as multiplicative factors with respect to the nominal value $IWC_0$*

*(i.e., $10^0$ = IWC$_0$, $10^{-1}$ = 0.1xIWC$_0$, …).*" has been added to the caption of Fig.3 for a clearer understanding of the results.

**Equations (10) and (11)**: the radiation being reflected by surface or atmosphere below the aircraft may transmitted through clouds. Why is the extinction of clouds ignored? Which is to say, instead of $Is(\lambda)R(\lambda)$, it may be more realistic using $Is(\lambda)R(\lambda)T(\lambda)$, T is the transmissivity.

We are aware of the limitation of not considering lower clouds in the model (as stated in line 262). However, our choice has been validated by the CPL data, which showed very little presence of lower clouds for the portion of the flight considered. We agree that Eq. (10) and Eq. (11) are simplifications. However, we would like to point out that the purpose of the equations in Section 3.1 is to provide a conceptual model to explain the mechanism of limb measurements of stratospheric cirrus clouds. The quantitative interpretation of the observations is performed using the RT model that can explicitly describe low altitude clouds.

**Line 379** '$Nice$ identifies the ice particle number density' I think it is called nominal ice concentration in Line 273, which should be mass concentration, not particle number density. Again, please keep the terms and symbols consistent if you are meaning the same thing.

We corrected this point here as well, following the previous comments regarding the ambiguity of these terms.

**Equation 13**: I think $\tau$=ln [$IobsIS$], based on Equation 7. Then Equation 13 misses a negative sign.

We thank the reviewer for this comment. Eq. (13) represents the generic expression for the optical depth (as stated in line 382) from which the different cases are derived, and it is expressed as an absolute value. We agree with the reviewer about the ambiguity of the sign; this has been resolved through introducing the minus symbol for the derived optical depth in Eq. (13), Eq. (15), Eq. (17), Eq. (18), and Eq. (20).

**Table 4**: size distribution 'bimodal 3-5 um', change as 'bimodal peak 3-5 um' might be clearer.

We thank the reviewer for the suggestion. This has been changed in the text of the caption of Table 4.

**Line 446**: 'the average ice particle number' is it the ice particle mass concentration as mentioned in Line 273? Note that particle number and particle mass are two different terms.

The reviewer is correct. It refers to mass concentration. The sentence has been modified in line 442 as:

"*The average ice particle mass concentration*".

**Line 447- 448**: 'SIWP and number concentration are proportional', be sure if you mean the number concentration or mass concentration.

The reviewer is correct again, this refers to mass concentration. This sentence has been modified in line 444 as:

"*SIWP and mass concentration*"

**Line 460**: 'This difference is likely due to an actual IWP higher than the modeled IWP'. This is confusing. SIWP? Ice concentrations?

The sentence has been rephrased as: "*This difference is likely due to an actual ice mass concentration higher than the modeled value.*" in line 454 to clarify this point.

**Line 463**: 'the model favors an ice density', is it the observation favors an ice density of about half of the nominal value? I think the nominal values is used in the model.

We confirm that the nominal value is used in the model. The sentence refers to the 7:26–7:40am portion of the flight only, where the simulation with half of the ice mass concentration is more in agreement with the observed measurements.

**Line 470-480**: Please explain more specifically how to get IWC from SIWP? Is there any equations or references to point the readers to a better understanding of the methods?

To improve the understanding of the methodology used to retrieve IWC from SIWP the following paragraph has been added in lines 463–468:

"*Based on the knowledge gained from our sensitivity calculations, we then used a linear interpolation of the modeled SIWP to the observed SIWP, in order to determine the IWC corresponding to the mini-DOAS measurements. The IWC for every observation is thus evaluated by scaling the nominal ice water content $IWC_0$, in order to achieve the best fit between the observed and modeled SIWP. The retrieved IWC varies between 0.001 gr $m^{-3}$ and 0.0025 gr $m^{-3}$ and is independent of the variability imposed by RT effects on*

*SIWP, these values are consistent with typical cirrus cloud ice water content levels observed in the TTL [Thornberry2017, Schiller2008]."*

**Line 472**: 'Four different cases were chosen…' please add details (date, location, flight altitude etc.) about these four cases.

We thank the reviewer for the suggestion. The four cases were chosen during Science Flight 2 in February 2014. The following paragraph has been added in lines 471–476:

"*Four different cases were chosen to represent the variability of the retrieved IWC from the measured SIWP. The four cases were chosen within the same Science Flight 2, recorded between the 16th and the 17th of February 2014, for the same geographical location (13°N,144°E). The four cases had similar altitudes (16.6 km) but different solar zenith (57.5°, 56.5°, 25°, 30.5°) and viewing azimuth (53.5°, 138.9°, 343.8°, 138.4°) angles. IWC for these four cases range from $5.4×10^{-3}$ g $m^{-3}$ to 0.02 g $m^{-3}$, capturing a wide range of cirrus cloud conditions*".

**Line 473**: 'retrieved IWCs for these four cases range from 5.4x10^-3 g/m3…', how well is this limit compared to lidar?

We chose to focus on the IWC from the in-situ instruments since a comparison of mini-DOAS with LIDAR data is challenging due to the different viewing strategies, i.e., limb vs nadir. The in-situ IWC observations are also more sensitive. We agree with the reviewer that a more formal intercomparison of the various methods would be advantageous. However, this was outside the scope of this study.

**Section 4.3**: the whole section discusses IWC rather than IWP.

The discussion of the interpretation of the results (related to SIWP, IWP, and IWC) starts at Section 4.2, following on to the next section. We kept IWP in the title only for consistency with the previous section. We agree with the reviewer's suggestion, and we renamed the section title in line 448 as: "*IWC retrievals*".

**Figure 8**, panel b, the unit of IWC 'gr m-3' → g m-3 to keep it consistent in the paper.

We thank the reviewer for spotting this typo. The label has been corrected.

**Line 512**: Fig. 8 shows SIWP not 'retrieved IWP'. Do you mean IWC?

"Shown" refers to the portion of the flight and not the IWP. To avoid confusion, the sentence has been modified in lines 514–515 as follows:

*"In order to evaluate the uncertainties introduced by a change in the RT input parameters to the retrieved IWC for the flight segment considered, we performed sensitivity tests for different ice cloud heights and particle size distributions"*.

**Lines 521-525**: It is confusing how to get IWPs and their uncertainties in this section. What is the best retrieval? How do you get the percentage uncertainty?

The following paragraph has been added in lines 525–529 to provide a better understanding of the IWP uncertainties in the retrieval:

*"IWP can be obtained from the simulated SIWP, which provides IWC that is then integrated over the known cloud extension (definition of IWP). In order to get the IWPs and their uncertainties , we ran the RT model starting with the nominal case (assumed as the best retrieval), defined by a cloud deck from 14–18 km, and then modifying the extension of the cloud and the azimuth angle by fixed amounts. IWP uncertainty is then expressed as a percentage of the variation of the retrieved IWP from the nominal case, for a variation of a specific parameter in the RT model (i.e., cloud height, SZA)."*

**Line 538**: 'as IWP is the IWC integrated over the vertical extent' move the definition of IWP before the discussion of IWP uncertainty and explain a little bit on why discuss IWP not IWC.

As mentioned in the reply to the previous comment, ice water path IWP is first defined in Section 2.2 (line 159), and we repeated the definition here in order to strengthen the point we were making. However, we did move the IWP definition before the discussion of the IWP uncertainties, as suggested by the reviewer, adding the following sentence in lines 525–527:

*"As IWP is the IWC integrated over the vertical extent of the cloud, we discuss IWP uncertainties rather than IWC uncertainties. In addition, IWP is related more closely to the variation of the cloud extent, as modeled in this sensitivity test."*

**Line 573**: 'lower detection sensitivity for IWC in limb geometry'. Do you mean higher detection sensitivity? Compare to what other approaches to demonstrate a better sensitivity of your approach?

The point of this sentence was mostly to remark on the use of limb scanning near-IR measurements as a tool to infer information on the IWC and the detection of subvisible cirrus cloud. The sentence has been rephrased in lines 576–578 as follows:

*"Finally, this approach has showed the ability to detect IWC in limb geometry, and its potential use as alternative method for the detection of cirrus at low ice water concentration."*

---

## Author Response (AR2)

**Investigation of cirrus cloud properties in the Tropical Tropopause Layer using high-altitude limb scanning near-IR spectroscopy during the NASA-ATTREX Experiment**

Santo Fedele Colosimo[1], Nathaniel Brockway[1], Vijay Natraj[2], Robert Spurr[3], Klaus Pfeilsticker[4], Lisa Scalone[5], Max Spolaor[1], Sarah Woods[6], and Jochen Stutz[1]

[1] Department of Atmospheric and Oceanic Sciences, UCLA, Los Angeles, CA, USA
[2] Jet Propulsion Laboratory, Caltech, Pasadena, CA, USA
[3] RT Solutions Inc., Cambridge, MA 02138, USA
[4] Institute of Environmental Physics, University of Heidelberg, Heidelberg, Germany
[5] Springer Heidelberg, Heidelberg, Germany
[6] Stratton Park Engineering Inc., Boulder, CO, USA

NOTE FROM THE AUTHORS

We would like to thank the anonymous reviewer #1, for the additional comments. The main text of the manuscript was modified following the reviewers' comments. We have provided a manuscript version with all the changes highlighted.

Please note that the title of the paper has been changed from:
*"Investigation of cirrus **clouds** properties in the Tropical Tropopause Layer using high-altitude limb scanning near-IR spectroscopy during the NASA-ATTREX Experiment"*
to:
*"Investigation of cirrus **cloud** properties in the Tropical Tropopause Layer using high-altitude limb scanning near-IR spectroscopy during the NASA-ATTREX Experiment"*
as it seems grammatically more correct the use of the word "*cloud*" instead of "*clouds*" in this context.

In the response to the reviewer, we addressed the questions/comments, pointing to specific changes in the manuscript.

Please see the uploaded version of the manuscript with the highlighted text for the changes made, and the following response to Reviewer #1 - Report #2.

**Authors' response to comments from Anonymous Referee #1 – Report #2**

**Line 269-270** Any reason the 2DS measurements were not used to calculate the ice particle optical properties?

As explained in one of our previous answers to a similar comment, we are aware of the limitations of the use of the Hawkeye-FCDP data. This instrument only measures particles between about 1–50 microns, relying on the Hawkeye-2D-S to cover a larger size range (10 microns to a few mm). We performed sensitivity studies and found that inclusion of larger particles in the radiative transfer models yielded unrealistic results. We therefore believe that these particles may not have been present at large concentrations in the air volume observed by our instrument. We agree that a more accurate description of the particle size distribution would have been beneficial for the presented methodology, but we consider that constraining the size range of the in-situ microphysical data for more direct comparison with the SIWP retrievals to be acceptable, given the scope of the paper.

**Line 478** gr>g

We thank the reviewer for spotting this typo. The right format for the physical units has been corrected accordingly.